# COMBINING Q-LEARNING AND SEARCH WITH AMORTIZED VALUE ESTIMATES

**Jessica B. Hamrick**
DeepMind
jhamrick@google.com

**Victor Bapst**
DeepMind
vbapst@google.com

**Alvaro Sanchez-Gonzalez**
DeepMind
alvarosg@google.com

**Tobias Pfaff**
DeepMind
tpfaff@google.com

**Théophane Weber**
DeepMind
theophane@google.com

**Lars Buesing**
DeepMind
lbuesing@google.com

**Peter W. Battaglia**
DeepMind
peterbattaglia@google.com

## ABSTRACT

We introduce "Search with Amortized Value Estimates" (SAVE), an approach for combining model-free Q-learning with model-based Monte-Carlo Tree Search (MCTS). In SAVE, a learned prior over state-action values is used to guide MCTS, which estimates an improved set of state-action values. The new Q-estimates are then used in combination with real experience to update the prior. This effectively *amortizes* the value computation performed by MCTS, resulting in a cooperative relationship between model-free learning and model-based search. SAVE can be implemented on top of any Q-learning agent with access to a model, which we demonstrate by incorporating it into agents that perform challenging physical reasoning tasks and Atari. SAVE consistently achieves higher rewards with fewer training steps, and—in contrast to typical model-based search approaches—yields strong performance with very small search budgets. By combining real experience with information computed during search, SAVE demonstrates that it is possible to improve on both the performance of model-free learning and the computational cost of planning.

## 1 INTRODUCTION

Model-based methods have been at the heart of reinforcement learning (RL) since its inception (Bellman, 1957), and have recently seen a resurgence in the era of deep learning, with powerful function approximators inspiring a variety of effective new approaches (Silver et al., 2018; Chua et al., 2018; Hamrick, 2019; Wang et al., 2019). Despite the success of model-free RL in reaching state-of-the-art performance in challenging domains (e.g. Kapturowski et al., 2018; Haarnoja et al., 2018), model-based methods hold the promise of allowing agents to more flexibly adapt to new situations and efficiently reason about what will happen to avoid potentially bad outcomes. The two key components of any such system are the *model*, which captures the dynamics of the world, and the *planning algorithm*, which chooses what computations to perform with the model in order to produce a decision or action (Sutton & Barto, 2018).

Much recent work on model-based RL places an emphasis on model learning rather than planning, typically using generic off-the-shelf planners like Monte-Carlo rollouts or search (see Hamrick (2019); Wang et al. (2019) for recent surveys). Yet, with most generic planners, even a perfect model of the world may require large amounts of computation to be effective in high-dimensional, sparse reward settings. For example, recent methods which use Monte-Carlo Tree Search (MCTS) require 100s or 1000s of model evaluations per action during training, and even upwards of a million simulations per time step at test time (Anthony et al., 2017; Silver et al., 2018). These large search budgets are required, in part, because much of the computation performed during planning—such

as the estimation of action values—is coarsely summarized in behavioral traces such as visit counts (Anthony et al., 2017; Silver et al., 2018), or discarded entirely after an action is selected (Bapst et al., 2019; Azizzadenesheli et al., 2018). However, large search budgets are a luxury that is not always available: many real-world simulators are expensive and may only be feasible to query a handful of times. In this paper, we explore preserving the value estimates that were computed by search by amortizing them via a neural network and then using this network to guide future search, resulting in an approach which works well even with very small search budgets.

We propose a new method called "Search with Amortized Value Estimates" (SAVE) which uses a combination of real experience as well as the results of past searches to improve overall performance and reduce planning cost. During training, SAVE uses MCTS to estimate the Q-values at encountered states. These Q-values are used along with real experience to fit a Q-function, thus amortizing the computation required to estimate values during search. The Q-function is then used as a prior for subsequent searches, resulting in a symbiotic relationship between model-free learning and MCTS. At test time, SAVE uses MCTS guided by the learned prior to produce effective behavior, even with very small search budgets and in environments with tens of thousands of possible actions per state—settings which are very challenging for traditional planners.

## 2 Background and Motivation

Unifying the complementary approaches of learning and search has been of interest to the RL and planning communities for many years (e.g. Gelly & Silver, 2007; Guo et al., 2014; Gu et al., 2016; Silver et al., 2016). SAVE is motivated in particular by two threads in this body of work: one which uses planning in-the-loop to produce experience for Q-learning, and one which learns a policy prior for guiding search. As we will describe next, both of these previous approaches can suffer from issues with training stability which are alleviated by SAVE by simultaneously using MCTS to strengthen an action-value function, and Q-learning to strengthen MCTS.

### 2.1 Learning from planned actions

A number of methods have explored learning from planned actions. Guo et al. (2014) trained a model-free policy to imitate the actions produced by an MCTS agent. Other methods use planning in-the-loop to recommend actions, which are then executed in the environment to gather experience for model-free learning (Silver et al., 2008; Gu et al., 2016; Azizzadenesheli et al., 2018; Shen et al., 2018; Lowrey et al., 2018; Bapst et al., 2019; Kartal et al., 2019). However, problems can arise when learning with actions that were produced via planning, even with off-policy algorithms like Q-learning. As noted by both Gu et al. (2016) and Azizzadenesheli et al. (2018), planning avoids suboptimal actions, resulting in a highly biased action distribution consisting of mostly good actions; information about suboptimal actions therefore does not get propagated back to the Q-function. As an example, consider the case where a Q-function recommends taking action $a$. During planning, this action is explored and is found to yield lower reward than expected. The planner will end up recommending some other action $a'$, which is executed in the environment and later used to update the Q-function. However, this means that the original action $a$ is never actually experienced and thus is never downweighed in the Q-function, resulting in poorly approximated Q-values.

One way to deal with this problem is to use a mixture of both on-policy and planned actions (Gu et al., 2016). However, this throws away information about poor actions which is acquired during the planning process. In SAVE, we instead make use of this information by using the values estimated during search to help fit the Q-function. If the search finds that a particular action is worse than previously thought, this information will be reflected by the estimated values and will thus ultimately get propagated back to the Q-function. We explicitly test and confirm this hypothesis in Section 4.2.

### 2.2 Using prior knowledge in search

Much research has leveraged prior knowledge in the context of MCTS (Gelly & Silver, 2007; 2011; Silver et al., 2016; Segler et al., 2018; Silver et al., 2017b; 2018; Anthony et al., 2017; 2019). Some of the most successful methods (Anthony et al., 2017; Silver et al., 2018) use a prior policy to guide search, the results of which are used to further improve the policy. However, such methods use information about past *behavior* to learn a policy prior—namely, the visit counts of actions during

search—and discard other search information such as inferred Q-values. We might anticipate one potential failure mode of such "count-based policy learning" approaches. Consider an environment with sparse rewards, where most actions are highly suboptimal. In the limit of infinite search, actions which have highest value will be visited most frequently, resulting in a policy that guides search towards regions of high value. However, in the regime of small search budgets, the search may very well end up exploring mostly suboptimal actions. These actions have higher visit counts, and so are reinforced, leading to the agent being *more* likely to explore poor actions.

Rather than implicitly biasing search towards value through the use of visit counts, SAVE relies on a prior that explicitly encodes knowledge about value. If SAVE ends up searching poor actions, it will learn that they have low values and this knowledge will be reflected in future searches. Thus, in contrast to count-based approaches, a SAVE agent will be less likely to visit poor actions in the future despite having frequently visited them in the past. We explicitly test and confirm this hypothesis in Section 4.1.

## 2.3 OTHER RELATED WORK

Finding effective ways of combining model-based and model-free experience has been of interest to the RL community for decades. Most famously, the Dyna algorithm (Sutton, 1990) proposes using real experience to learn a model and then using the model to train a model-free policy. A number of more recent works have explored how to incorporate this idea into deep architectures (Kalweit & Boedecker, 2017; Feinberg et al., 2018; Buckman et al., 2018; Serban et al., 2018; Kurutach et al., 2018; Kaiser et al., 2019), with an emphasis on dealing with the errors that are introduced by approximate models. In these approaches, the policy or value function is typically trained using on-policy rollouts from the model without using additional planning. Another way to combine model-free and model-based approaches is "implicit planning", in which the computation of a planner is built into the architecture of a neural network itself (Weber et al., 2017; Buesing et al., 2018; Pascanu et al., 2017; Silver et al., 2017b; Oh et al., 2017; Guez et al., 2018; Farquhar et al., 2018; Hamrick et al., 2017; Srinivas et al., 2018; Yu et al., 2019; Tamar et al., 2016; Karkus et al., 2017). While SAVE is not an implicit planning method, it shares similarities with such methods in that it also tightly integrates planning and learning.

## 3 METHOD

SAVE features two main components (Figure 1). First, we use a search policy that incorporates the Q-function $Q_\theta(s, a)$ as a **prior over Q-values** that are estimated during search. Second, to train the Q-function we rely on an objective function that combines both the TD-error from Q-learning with an **amortization loss** that amortizes the value computation performed by the search. The amortization loss, combined with the prior over Q-values, thus enables future searches to build on previous ones, resulting in stronger search performance overall.

## 3.1 STANDARD MCTS

Before explaining how SAVE leverages search, we briefly describe the standard MCTS algorithm (Kocsis & Szepesvári, 2006; Coulom, 2006). While we focus here on the single-player setting, we note that the formulation of MCTS (and by extension, SAVE) is similar for two-player settings. MCTS uses a simulator or model of the environment to explore possible future states and actions, with the aim of finding

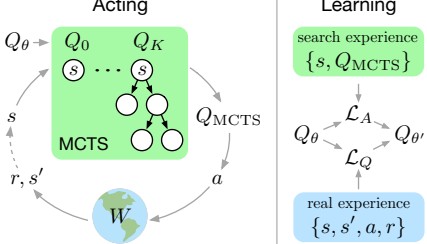

Figure 1: Illustration of SAVE. When acting, the agent uses a Q-function, $Q_\theta$, as a prior for the Q-values estimated during MCTS. Over $K$ steps of search, $Q_0 \equiv Q_\theta$ is built up to $Q_K$, which is returned as $Q_{\text{MCTS}}$ (Equations 1 and 4). From $Q_{\text{MCTS}}$, an action $a$ is selected via epsilon-greedy and the resulting experience $(s, a, r, s', Q_{\text{MCTS}})$ is added to a replay buffer. When learning, the agent uses real experience to update $Q_\theta$ via Q-learning ($\mathcal{L}_Q$) as well as an amortization loss ($\mathcal{L}_A$) which regresses $Q_\theta$ towards the Q-values estimated during search (Equation 6).

a good action to execute from the current state, $s_0$. In MCTS, we assume access to a budget of $K$ iterations (or simulations). The $k^{\text{th}}$ iteration of MCTS consists of three phases: selection, expan-

sion, and backup. In the selection phase, we expand a search tree beginning with the current state and taking actions according to a search policy:

$$\pi_k(s) = \arg\max_a \left( Q_k(s, a) + U_k(s, a) \right), \tag{1}$$

where $Q_k$ is the currently estimated value of taking action $a$ while in state $s$, which will be explained further below. $U_k(s, a)$ is the UCT exploration term:

$$U_k(s, a) = c_{\text{UCT}} \sqrt{\frac{\log\left(\sum_a N_k(s, a)\right)}{N_k(s, a)}}, \tag{2}$$

where $N_k(s, a)$ is the number of times we have explored taking action $a$ from state $s$ and $c_{\text{UCT}}$ is a constant that encourages exploration. This selection procedure is repeated for $T - 1$ times, until a new action $a_{T-1}$ that had not previously been explored is chosen from state $s_{T-1}$. This begins the expansion phase, during which $a_{T-1}$ is executed in the simulator, resulting in a reward $r_{T-1}$ and new state $s_T$. The new state $s_T$ is added to the search tree, and its value $V(s_T)$ is estimated either via a state-value function or (more traditionally) via a Monte-Carlo rollout. At this point the backup phase begins, during which the value of $s_T$ is used to update (or "back up") the values of its parent states earlier in the tree. Specifically, for state $s_t$, the $i^{\text{th}}$ backed up return is estimated as:

$$R_i(s_t, a_t) = \gamma^{T-t} V(s_T) + \sum_{j=t}^{T-1} \gamma^{j-t} r_j, \tag{3}$$

where $\gamma$ is the discount factor and $r_j$ was the reward obtained after executing $a_j$ in $s_j$ when traversing the search tree. These backups are then used to estimate the Q-function in Equation 1 as $Q_k(s, a) = \sum_{i=1}^{N_k(s,a)} R_i(s, a)/N_k(s, a)$.

## 3.2 Incorporating a prior during search

SAVE makes several changes to the standard MCTS procedure. First, it assumes it has visited every state and action pair once by initializing $N(s, a) = 1$ for all states and actions.[1] Second, for each of these state-action pairs, it assumes a prior estimate of its value, $Q_\theta(s, a)$, and uses this as an initial estimate for $Q_k$, similar to Gelly & Silver (2007; 2011):

$$Q_k(s, a) = \frac{Q_\theta(s, a) + \sum_{i=1}^{N_k(s,a)-1} R_i(s, a)}{N_k(s, a)}. \tag{4}$$

where $Q_0(s, a) := Q_\theta(s, a)$. Third, rather than using a separate state-value function or Monte-Carlo rollouts to estimate the value of new states, SAVE uses the same state-action value function, i.e. $V(s) := \max_a Q_\theta(s, a)$. These three changes provide a mechanism for incorporating Q-based prior knowledge into MCTS: specifically, SAVE acts as if it has visited every state-action pair once, with the estimated values being given by $Q_\theta$. Roughly speaking, this can be interpreted as using MCTS to perform Bayesian inference over Q-values, with the prior specified by $Q_\theta$ with a weight equivalent to a pseudocount of one. This set of changes contrasts with UCT, which does not incorporate prior knowledge, as well as PUCT (Rosin, 2011; Silver et al., 2017a; 2018), which incorporates prior knowledge via a policy in the exploration term $U_k(s, a)$.

After $K$ iterations, we return $Q_{\text{MCTS}}(s, a) := Q_K(s, a)$ and select an action to execute in the environment via epsilon-greedy over $Q_{\text{MCTS}}(s_0, a)$. After the action is executed, we store the resulting experience along with a copy of $Q_{\text{MCTS}}(s_0, \cdot) \equiv \{Q_{\text{MCTS}}(s_0, a_i)\}_i$ in the replay buffer. This process is illustrated in Figure 1 (left).

## 3.3 Q-learning with an amortization loss

During learning, the results of the search are amortized into an updated prior $Q_{\theta'}$ (Figure 1, right). We impose an amortization loss $\mathcal{L}_A$ which encourages the distribution of Q-values output by the neural network to be similar to those estimated by MCTS. The amortization loss is defined to be

---

[1] We could also consider initializing $N(s, a)$ based on an estimate of previous visit counts, which we leave as an interesting direction for future work.

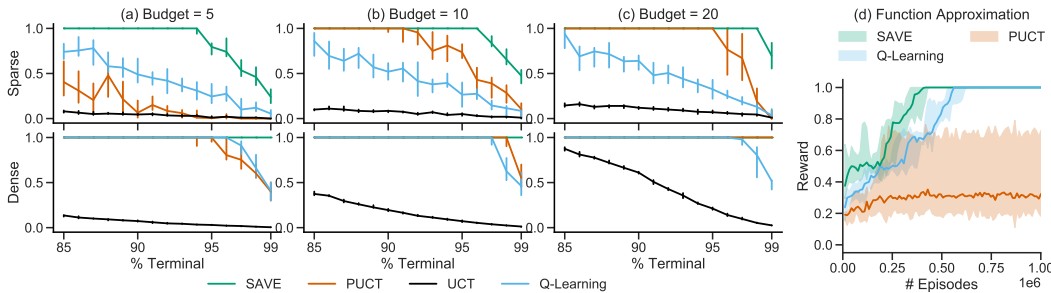

Figure 2: Results on Tightrope. (a-c) Tabular results comparing SAVE, PUCT, UCT, and Q-learning (with MCTS at test time) for varying percentages of terminal actions on the $x$-axes and for different search budgets. The $y$-axes show reward for either the sparse or dense reward setting of Tightrope. Lines show medians across 20 seeds, with error bars showing 95% confidence intervals. (d) Results on Tightrope when using function approximation, comparing SAVE with PUCT and Q-learning. Lines show medians across 10 seeds, with shaded regions indicating min and max seeds.

the cross-entropy between the softmax of the Q-values before ($Q_\theta$) and after ($Q_{\mathrm{MCTS}}$) MCTS. This cross-entropy loss achieves better performance than alternatives like L2, as described in Section 4.2. Setting $\mathbf{p}_{\mathrm{MCTS}} = \mathrm{softmax}_\tau(Q_{\mathrm{MCTS}}(s, \cdot))$ and $\mathbf{p}_\theta = \mathrm{softmax}_\tau(Q_\theta(s, \cdot))$, where $\tau = 1$ is the softmax temperature, the loss is defined as:

$$\mathcal{L}_A(\theta, \mathcal{D}) = -\frac{1}{N} \sum_{\mathcal{D}} (\mathbf{p}_{\mathrm{MCTS}})^\top \log \mathbf{p}_\theta, \tag{5}$$

where $\mathcal{D}$ is a batch of $N$ experience tuples $(s_t, a_t, r_t, s_{t+1}, Q_{\mathrm{MCTS}}(s_t, \cdot))$ sampled from the replay buffer. This amortization loss is linearly combined with a Q-learning loss,

$$\mathcal{L}(\theta, \mathcal{D}) = \beta_Q \mathcal{L}_Q(\theta, \mathcal{D}) + \beta_A \mathcal{L}_A(\theta, \mathcal{D}), \tag{6}$$

where $\beta_Q$ and $\beta_A$ are coefficients to scale the loss terms. $\mathcal{L}_Q$ may be any value-based loss function, such as that based on 1-step TD targets, $n$-step TD targets, or $\lambda$-returns (Sutton, 1988). The amortization loss does make SAVE more sensitive to off-policy experience, as the values of $Q_{\mathrm{MCTS}}$ stored in the replay buffer will become less useful and potentially misleading as $Q_\theta$ improves; however, we did not find this to be an issue in practice.

## 4 EXPERIMENTS

We evaluated SAVE in four distinct settings that vary in their branching factor, sparsity of rewards, and episode length. First, we demonstrate through a new Tightrope environment that SAVE performs well in settings where count-based policy approaches struggle, as discussed in Section 2.2. Next, we show that SAVE scales to the challenging Construction domain (Bapst et al., 2019) and that it alleviates the problem with off-policy actions discussed in Section 2.1. We also perform several ablations to tease apart the details of SAVE. Finally, we demonstrate that SAVE dramatically improves over Q-learning in a new and even more difficult construction task called Marble Run, as well as in more standard environments like Atari (Bellemare et al., 2013). In all our experiments we use SAVE with a perfect model of the environment, though we expect our approach would work with learned models as well.

### 4.1 TIGHTROPE

In Section 2.2, we hypothesized that approaches which use count-based policy learning rather than value-based learning (e.g. Anthony et al., 2017; Silver et al., 2018) may suffer in environments with large branching factors, many suboptimal actions, and small search budgets. To test this hypothesis, we developed a toy environment called *Tightrope* with these characteristics. Tightrope is a deterministic MDP consisting of 11 labeled states linked together in a chain. At each state, there are 100 actions to take, $M\%$ of which are terminal (meaning that when taken they cause the episode to end).

The other non-terminal actions will cause the state to transition to the next state in the chain. We considered two settings of the reward function: *dense* rewards, in which case the agent receives a reward of 0.1 when making it to the next state in the chain and 0 otherwise; and *sparse* rewards, in which case the agent receives a reward of 1 only when making it to the final state. In the sparse reward setting, we randomly selected one state in the chain to be the "final" state to form a curriculum over the length of the chain. With the exception of the final state in the sparse reward setting, the transition function of the MDP is exactly the same across episodes, with the same actions always having the same behavior.

**Tabular Results**   We first examined the behavior of SAVE on Tightrope in a tabular setting to eliminate potential concerns about function approximation (see Section B.2). We compared SAVE to three other agents. **UCT** is a pure-search agent which runs MCTS using a UCT search policy with no prior. It uses Monte-Carlo rollouts following a random policy to estimate $V(s)$. **PUCT** is based on AlphaZero (Silver et al., 2018) and uses a policy prior (which is learned from visit counts during MCTS) and state-value function (which is learned from Monte-Carlo returns). During search, the policy is used in the PUCT exploration term and the value function is used for bootstrapping. More details on PUCT in general are provided in Section A.3. **Q-Learning** performs one-step tabular Q-learning during training, and MCTS at test time using the same search procedure as SAVE.

Figure 2a-c illustrates the results in the tabular setting after 500 episodes. UCT, which does not use any learning, illustrates the difficulty of using brute-force search. Q-learning, which does not use any search during training, is slow to converge to a solution within the 500 episodes, particularly in the sparse reward setting; additionally, adding search at test time does not substantially improve things. Although the incorporation of learning with PUCT does improve the results, we can see that with small search budgets and high proportions of terminal actions, PUCT struggles to remember which actions are safe (nonterminal), especially in the sparse reward setting. In contrast, SAVE solves the Tightrope environment in all of the dense reward settings and most of the sparse reward settings. As the search budget increases, we see that both PUCT and SAVE reliably converge to a solution; thus, if a large search budget is available both methods may fare equally well. However, if only a small search budget is available, SAVE results in much more reliable performance.

**Function Approximation Results**   We also looked at the ability of SAVE and PUCT to solve the Tightrope environment when using function approximation, along with a model-free Q-learning baseline (see Section B.3). We evaluated all agents on the sparse reward version of Tightrope with 95% terminal actions, and used a search budget of 10 (except for Q-learning, which used a test budget of zero). The results, shown in Figure 2d, follow the same pattern as in the tabular setting.

## 4.2   CONSTRUCTION

We next evaluated SAVE in three of the Construction tasks explored by Bapst et al. (2019), in which the goal is to stack blocks to achieve a functional objective while avoiding collisions with obstacles. In *Connecting*, the goal is to connect a target point in the sky to the floor. In *Covering*, the goal is to cover obstacles from above without touching them. *Covering Hard* is the same as Covering, except that only a limited number of blocks may be used. The Construction tasks are challenging for model-free approaches because there is a combinatorial space of possible scenes and the physical dynamics are challenging to predict. However, they are also difficult for traditional search methods, as they have huge branching factors with up to tens of thousands of possible actions per state. Additionally, the simulator in the Construction tasks is expensive to query, making it infeasible to use with search budgets of more than 10-20.

To implement SAVE, we used the same agent architecture as Bapst et al. (2019). We compared SAVE to a baseline version of **SAVE without amortization loss** (i.e., $\mathcal{L}(\theta, \mathcal{D}) = \beta_Q \mathcal{L}_Q(\theta, \mathcal{D})$), similar to the MCTS agent described in Bapst et al. (2019). We also compared to a **Q-learning** baseline which performs pure model-free learning during training (but which may also utilize MCTS at test time using the same search procedure as SAVE), as well as a **UCT** baseline which did not use any learning (but which did use a pretrained value function for bootstrapping). For SAVE-based agents, we used a training budget of 10 simulations and varied the budget at test time; for UCT, we used a constant budget of 1000 simulations at test time (see Appendix C).

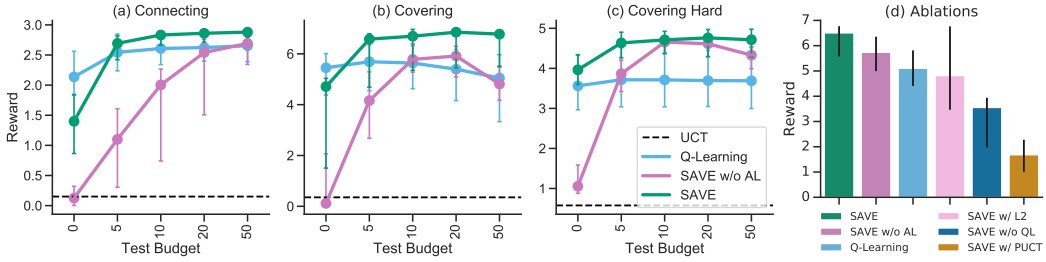

Figure 3: Results on Construction. (a-c) Each subplot shows results for SAVE, SAVE without amortization loss, Q-learning with MCTS at test time, and pure search (UCT). The $x$-axis shows the effect of increasing the number of MCTS simulations at test time. During training, SAVE with and without amortization loss used a search budget of 10 simulations. UCT used a search budget of 1000 simulations. Points show medians across 10 seeds, and error bars indicate min and max seeds. (d) Ablation experiments on the Covering task. We compare SAVE to variants that do not have an amortization loss, which use an L2 amortization loss, which do not use the Q-Learning loss, and which use PUCT rather than UCT. Results are shown at the hardest level of difficulty for the Covering task with a test budget of 10. The colored bars show median reward across 10 seeds, and error bars show min and max seed.

**Results** Figure 3a-c shows the results on the three construction tasks. The poor performance of UCT (dotted lines) highlights the need for prior knowledge to manage the huge branching factor in these domains. While model-free Q-learning improves performance, simply performing search on top of the learned Q-values only results in small gains in performance, if any. The performance of SAVE without amortization loss highlights exactly the issue discussed in Section 2.1. Without the amortization loss, the Q-learning component of SAVE only learns about actions which have been selected via search, and thus rarely sees highly suboptimal actions, resulting in a poorly approximated Q-function. Indeed, as we can see in the case where the search budget is zero, the agent's performance falls off dramatically, suggesting that the underlying Q-values are poor. Using search at test time can make up for this problem to some degree, but only when used with a budget very close to that with which it was trained: large search budgets can actually result in *worse* search performance (e.g. in Covering and Covering Hard) because the poor Q-values are also being used for bootstrapping during the search. It is only by leveraging search during training time and incorporating an amortization loss do we see a synergistic result: using SAVE results in higher rewards across all tasks, strongly outperforming the other agents.

**Ablation Experiments** In the past two sections, we compared SAVE to alternatives which do not include an amortization loss, or which use count-based policy learning rather than value-based learning. However, a number of additional questions remain regarding the architectural choices in SAVE. To address these, we ran a number of ablation experiments on the Covering task, with the results shown in Figure 3d. Specifically, we compared SAVE with versions that use an L2 loss (rather than cross entropy), that do not use the Q-learning loss, and that use the Q-values to guide search via PUCT rather than initializing $Q_0$. Overall, we find that the choices made in SAVE result in the highest levels of performance. Of particular note is the ablation that uses the L2 loss, indicating that the softmax cross entropy loss plays an important role in SAVE's performance. We speculate this is true for two reasons. First, because we use small search budgets, the estimated $Q_{\mathrm{MCTS}}$ is likely to be noisy, and thus it may be more robust to preserve just the relative magnitudes of action values rather than exact quantities. Second, the cross entropy loss means that $Q_\theta$ need not represent the values of poor actions exactly, thus freeing up capacity in the neural network to more precisely represent the values of good actions. Details and further discussion is provided in Section C.3. We also compared to a policy-based PUCT agent like that described in Section 4.1, but found this did not achieve positive reward on the harder tasks like Covering. This result again highlights the same problem with count-based policy training and small search budgets, as discussed in Section 2.2.

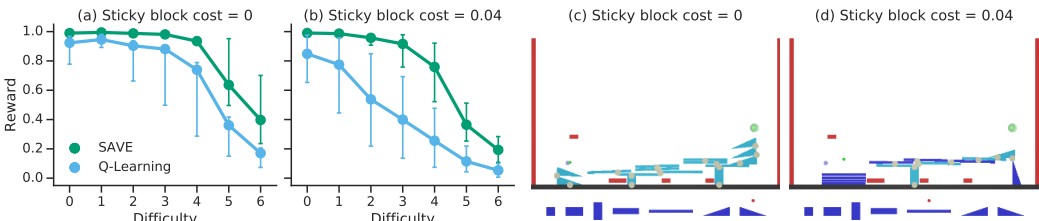

Figure 4: (a-b) Results on the Marble Run environment for model-free Q-Learning as well as SAVE as a function of curriculum difficulty level, for two different settings of the cost of "sticky" blocks. Points indicate medians across 10 seeds, and error bars show min and max seeds. (c-d) Structures built by SAVE which solve the same scene for two different costs of sticky blocks (difficulty 6). Additional videos showing agent behavior are available at `https://tinyurl.com/yxm4ma47`.

### 4.3 MARBLE RUN

SAVE is able to achieve near-ceiling levels of performance on the original Construction tasks. Thus, we developed a new task in the style of the previous Construction tasks called *Marble Run* which is even more challenging in that it involves sparser rewards and a more complex reward function. Specifically, the goal in Marble Run is to stack blocks to enable a marble to get from its original starting position to a goal location, while avoiding obstacles. At each step, the agent may choose from a number of differently shaped rectangular blocks as well as ramp shapes, and may choose to make these blocks "sticky" (for a price) so that they stick to other objects in the scene. The episode ends once the agent has created a structure that would get the marble to the goal. The agent receives a reward of one if it solves the scene, and zero otherwise.

We used the same agent architecture and training setup as with the Construction tasks, except for the curriculum. Specifically, we found it was important to train agents on this task using an adaptive curriculum over difficulty levels rather than a fixed linear curriculum. Under the adaptive curriculum, we only allowed an agent to progress to the next level of difficulty after it was able to solve at least 50% of the scenes at the current level of difficulty. Further details of the Marble Run task and the curriculum are given in Appendix D.

**Results**  Figure 4 shows the results for SAVE and Q-learning for the two different costs of sticky blocks, as as well as some example constructions. SAVE progresses more quickly through the curriculum and reaches higher levels of difficulty (see Figure D.1) and overall achieves much higher levels of reward at every difficulty level. Additionally, we found that the Q-learning agent reliably becomes unstable and collapses at around difficulty 4-5 (see Figure D.2), while SAVE does not have this problem. Qualitatively (Figure 4c-d), SAVE is able to build structures which allow the marble to reach targets that are raised above the floor while also spanning multiple obstacles.

These results on Marble Run also allow us to address the trade-off between model-free experience versus planned experience. Specifically, with a search budget of 10, SAVE effectively sees 10 times as many transitions as a model-free agent trained on the same number of environment interactions. Would a model-free agent trained for 10 times as long achieve equivalent performance? As can be seen in Figure D.2, this is not the case: the model-free agent sees more episodes but results in worse performance. We find the same result in other Construction tasks as well (see Section C.4). This highlights the positive interaction that occurs when learning both from experience generated from planned actions and from the values estimated during search.

### 4.4 ATARI

To demonstrate that SAVE is applicable to more standard environments, we also evaluated it on a subset of Atari games (Bellemare et al., 2013). We implemented SAVE on top of R2D2, a distributed Q-learning agent that achieves state-of-the-art results on Atari (Kapturowski et al., 2018). To allow

for a fair comparison[2] between purely model-free R2D2 and a version with SAVE, we controlled R2D2 to have the same replay ratio as SAVE and then tuned its hyperparameters to have approximately the same level of performance as the baseline version of R2D2 (see Appendix E). We find that SAVE outperforms or equals this controlled version of R2D2 in all games, with particularly high performance on Frostbite, Alien, and Zaxxon (shown in Figure 5). SAVE also outperforms the baseline version of R2D2 (see Table E.1 and Figure E.1).

## 5 DISCUSSION

We introduced SAVE, a method for combining model-free Q-learning with MCTS. During training, SAVE leverages MCTS to infer a set of Q-values, and then uses a combination of real experience plus the estimated Q-values to fit a Q-function, thus *amortizing* the value computation of previous searches via a neural network. The Q-function is used as a prior to guide future searches, enabling even stronger search performance, which in turn is further amortized via the Q-function. At test time, SAVE can be used to achieve high levels of reward with only very small search budgets, which we demonstrate across four distinct domains: Tightrope, Construction (Bapst et al., 2019), Marble Run, and Atari (Bellemare et al., 2013; Kapturowski et al., 2018). These results suggest that SAVEing the experience generated by search in an explicit Q-function, and initializing future searches with that information, offers important advantages for model-based RL.

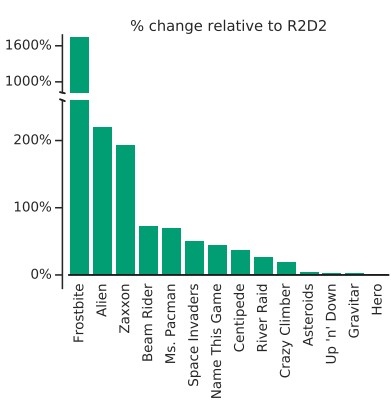

Figure 5: Results on Atari.

When combining Q-values estimated both from prior searches and real experience, it may also be useful to account for the quality or confidence of the estimated Q-values. Count-based policy methods (Anthony et al., 2017; Silver et al., 2018) do this by leveraging an estimate of confidence based on visit counts: actions with high visit counts should both have high value (or else they would not have been visited so much) and high confidence (because they have been explored extensively). However, as we have shown, relying solely on visit counts can result in poor performance when using small search budgets (Section 4.1). A key future direction will be to amortize *both* the computation of value and of reliability, achieving the best of both SAVE and count-based methods. Encoding confidence estimates into the Q-values may also be helpful for applying SAVE to settings with learned models, which may have non-trivial approximation errors. In particular, it may be helpful to attenuate the contribution of search-estimated Q-values to the Q-prior both when an action has not been sufficiently explored and when model error is high.

Our work demonstrates the value of amortizing the Q-estimates that are generated during MCTS. Indeed, we have shown that by doing so, SAVE reaches higher levels of performance than model-free approaches while using less computation than is required by other model-based methods. More broadly, we suggest that SAVE can be interpreted as a framework for ensuring that the valuable computation performed during search is preserved, rather than being used only for the immediate action or summarized indirectly via frequency statistics of the search policy. By following this philosophy and tightly integrating planning and learning, we expect that even more powerful hybrid approaches can be achieved.

## 6 ACKNOWLEDGEMENTS

We would like to thank GB Parascandolo, George Papamakarios, Nicolas Heess, Ioannis Antonoglou, Thomas Hubert, Julian Schrittweiser, and David Silver for helpful comments and feedback on this project.

---

[2]R2D2 is very sensitive to the speed of the learners and actors. If the actors are slower (which they will be when performing search), the replay ratio will increase which thus affects performance.

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

# A  FURTHER AGENT DETAILS

In all experiments except Tabular Tightrope (see Section B.2) and Atari (see Appendix E), we use a distributed training setup with 1 GPU learner and 64 CPU actors. Our setup was implemented using TensorFlow (Abadi et al., 2016) and Sonnet (Reynolds et al., 2017), and gradient descent was performed using the Adam optimizer (Kingma & Ba, 2014) with the TensorFlow default parameter settings (except learning rate).

## A.1  Q-LEARNING

Except for in Atari (see Appendix E), we used a 1-step implementation of Q-learning, with the standard setup with experience replay and a target network (Mnih et al., 2015). We controlled the rate of experience processed by the learner such that the average number of times each transition was replayed (the "replay ratio") was kept constant. For all experiments, we used a batch size of 16, a learning rate of 0.0002, a replay size of 4000 transitions (with a minimum history of 100 transitions), a replay ratio of 4, and updated the target network every 100 learning steps.

We used a variant of epsilon-greedy exploration described by Bapst et al. (2019) in which epsilon is changed adaptively over the course of an episode such that it is lower earlier in the episode and higher later in the episode, with an average value of $\epsilon$ over the whole episode. We annealed the average value of $\epsilon$ from 1 to 0.01 over 1e4 episodes.

## A.2  SAVE

**Algorithm A.1** Pseudocode for the SAVE algorithm.

---
1: **procedure** SAVE($\theta$)
2:     **while** true **do**
3:         Begin episode at $s$
4:         **while** acting **do**
5:             Estimate $Q_{\mathrm{MCTS}}(s, \cdot) \leftarrow \mathrm{MCTS}(s, Q_\theta)$
6:             Select $a$ using epsilon-greedy from $Q_{\mathrm{MCTS}}(s, \cdot)$
7:             Execute $a$ in environment and receive $s', r$
8:             Add $(s, a, r, s', Q_{\mathrm{MCTS}}(s, \cdot))$ to replay buffer
9:             $s \leftarrow s'$
10:        **while** learning **do**
11:            Sample minibatch of experience from the replay buffer
12:            Update $\theta$ to minimize Equation 6
13:
14: **procedure** MCTS($s_0, Q_\theta$)
15:     $Q_0(s, a) \leftarrow Q_\theta(s, a)$ for all $s, a$
16:     $N_0(s, a) \leftarrow 1$ for all $s, a$
17:     $k \leftarrow 0$
18:     **while** search budget remains ($k < K$) **do**
19:         Traverse the search tree with $\pi_k$ (Equation 1)
20:         Expand new state $s_T$ and add it to the search tree
21:         Evaluate $\max_a Q_\theta(s_T, a)$ and backup returns (Equation 3)
22:         Set $N_{k+1}(s, a) \leftarrow N_k(s, a)$ and then increment counts of visited states and actions
23:         Compute estimates for $Q_{k+1}(s, a)$ (Equation 4)
24:         $k \leftarrow k + 1$
25:     Return $\{Q_{\mathrm{K}}(s_0, a_i)\}_i$

---

The SAVE agent is implemented as described in Section 3 and Algorithm A.1 provides additional pseudocode explaining the algorithm. In Algorithm A.1, we provide an example of using SAVE in an episode setting where learning happens after every episode; however, SAVE can be used in any Q-learning setup including in distributed setups where separate processes are concurrently acting and learning. In particular, in our experiments we use the distributed setup described in Section A.1. Note that when performing epsilon-greedy exploration (Line 6 of Algorithm A.1), we either choose

an action uniformly at random with probability $\epsilon$, and otherwise choose the action with the highest value of $Q_{\text{MCTS}}$ out of the actions which were explored during search (i.e., we do not consider actions that were not explored, even if they have a higher $Q_{\text{MCTS}}$). In all experiments (except tabular Tightrope), we use a UTC exploration constant of $c = 2$, though we have found SAVE's performance to be relatively robust to this parameter setting.

## A.3 PUCT

The PUCT search policy is based on that described by Silver et al. (2017a) and Silver et al. (2018). Specifically, we choose actions during search according to Equation 1, with:

$$Q_k = \frac{\sum_{i=1}^{N_k(s,a)} R_i(s,a)}{N_k(s,a)} \tag{7}$$

$$U_k(s,a) = c \cdot \pi(s,a) \frac{\sqrt{\sum_a N_k(s,a)}}{N_k(s,a) + 1}$$

where $c$ is an exploration constant, $\pi(s,a)$ is the prior policy, and $N_k(s,a)$ is the total number of times action $a$ had been taken from state $s$ at iteration $k$ of the search. Like Silver et al. (2017a; 2018), we add Dirichlet noise to the prior policy:

$$\pi(s,a) = (1 - \epsilon) \cdot \pi_\theta(s,a) + \epsilon\eta,$$

where $\eta \sim \text{Dir}(1/n_{\text{actions}})$. In our experiments we set $\epsilon = 0.25$ and $c = 2$. During training, after search is complete, we sample an action to execute in the environment from $\pi_{\text{MCTS}}(s_0, a) = N_K(s_0, a)/\sum_a N_K(s_0, a)$. At test time, we select the action which has the maximum visit count (with random tie-breaking).

To train the PUCT agent, we used separate policy $\pi_\theta(s,a)$ and value $V_\theta(s)$ heads which were trained using a combined loss (Equation 6), with:

$$\mathcal{L}_Q = \frac{1}{N} \sum_{\mathcal{D}} \left\| V_\theta(s) - R \right\|_2$$

$$\mathcal{L}_A = -\frac{1}{N} \sum_{\mathcal{D}} \pi_{\text{MCTS}}(s, \cdot)^\top \log \pi_\theta(s, \cdot)$$

where $R$ is the Monte-Carlo return observed from state $s$. We used fixed values of $\beta_Q = 0.5$ and $\beta_A = 0.5$ in all our experiments with PUCT. We used the same replay and training setup as used in the Q-learning and SAVE agents, with two exceptions. First, we additionally include episodic Monte-Carlo returns $R$ and policies $\pi_{\text{MCTS}}$ in the replay buffer so they can be used during learning. Second, we did not use $\epsilon$-greedy exploration (because the Dirichlet noise in the PUCT term already enables sufficient exploration).

We tried several different hyperparameter settings and variants of the PUCT agent to attempt to improve the results. For example, we tried using a 1-step TD error for learning the values, which should have lower variance and thus result in more stable learning of values. We also tried reducing the replay ratio to 1 and the replay size to 400 in order to make the experience for training more on-policy. However, we did not find that these changes improved the results. We also tried different settings of $\epsilon$ for the Dirichlet noise, but found that lower values resulted in too little exploration, while higher values resulted in too much exploration.

## B   Details on Tightrope

### B.1   Environment

The Tightrope environment has 11 states which are connected together in a chain. Each state has 100 actions, $M\%$ of which will cause the episode to terminate when executed and the rest of which will cause the environment to transition to the next state. Each state is represented using a vector of 50 random values drawn from a standard normal distribution, which are the same across episodes. The indices of terminal actions are selected randomly and are different for each state but are consistent across episodes. Agents always begin in the first state of the chain.

In the sparse reward setting, we randomly select one of the states in the chain to be the "final" state (excluding the first state), to enable the agent to sometimes train on easy problems and sometimes train on hard problems. If the agent reaches this final state, it receives a reward of 1 and the episode terminates. If it takes a non-terminal action, it transitions to the next state in the chain and receives a reward of 0. Otherwise, if it takes a terminal action, the episode terminates and the agent receives a reward of 0.

In the dense reward setting, the "final" state is always chosen to be the last state in the chain. If the agent reaches the final state in the chain, it receives a reward of 0.1 and the episode terminates. If it takes a non-terminal action, it transitions to the next state in the chain and receives a reward of 0.1. Otherwise, if it takes a terminal action, the episode terminates with a reward of 0.

## B.2 TABULAR EXPERIMENTS

During training, we execute each tabular agent in the environment until the episode terminates. Then, we perform a learning step using the experience generated from the previous episode. This process repeats for some number of episodes (in our experiments, 500). After training, we execute each agent in the environment 100 times and compute the average reward achieved across these 100 episodes. For all cases in which search is used, we use a UCT exploration constant of $c = 0.1$.

**Q-Learning**  Tabular Q-learning begins with a table of state-action values initialized to zero. We perform epsilon-greedy exploration with $\epsilon = 0.1$, and add the resulting experience to a replay buffer with maximum size of 1000 transitions. We perform episodic learning, where during each episode the Q-values are fixed and after the episode is complete we update the Q-values by performing a single pass through the experience in the replay buffer in a random order. We use a learning rate of $\beta_Q = 0.01$. At test time, the Q-learning agent uses MCTS in the same manner as SAVE.

**SAVE**  Tabular SAVE begins with a table of state-action values initialized to zero. During search, values are looked up in this table and used to initialize $Q_0$. The values are also for bootstrapping. During learning, we perform both Q-learning (as described in the Q-learning agent) as well as an update based on the gradient of the cross-entropy amortization loss (Equation 6). We use $\beta_Q = 0.01$ and $\beta_A = 1$.

**PUCT**  Tabular PUCT begins with two tables; one with state values (initialized to zero) and one with action probabilities (initialized to the uniform distribution). During search, action probabilities are looked and used in the PUCT term, while state values are looked up and used for bootstrapping. Search proceeds as described in Section A.3. During learning, $\pi_{\mathrm{MCTS}}$ is copied back into the action probability table (this is equivalent to an L2 update with a learning rate of 1); we also experimented with doing an update based on the cross entropy loss but found this resulted in worse performance. The value at episode $t$ is given by:

$$V_t(s) = (1 - \alpha)V_{t-1}(s) + \alpha R_{t-1}(s), \tag{8}$$

where $R_{t-1}(s)$ is the return obtained after visiting state $s$ during episode $t - 1$. In our experiments we used $\alpha = 0.5$. We also experimented with using Q-learning rather than Monte-Carlo returns, but found that these resulted in similar levels of performance.

**UCT**  The UCT agent is as described in Section 3.1, with $V(s)$ at unexplored nodes estimated via a Monte-Carlo rollout under a uniform random policy. The only difference from regular UCT is that we did not require all actions to be visited before descending down the search tree; unvisited actions were initialized to a value of zero. For Tightrope, this is the optimal setting of the default Q-values because all possible rewards are greater than or equal to zero. Once an action is found with non-zero reward the best option is to stick with it, so it would not make sense to set the values optimistically. Actions that cause the episode to terminate have a reward of zero, so it would also not make sense to set the values pessimistically as this would lead to over-exploring terminal actions. Setting the values to the average of the parent would either have the effect of setting to zero or setting optimistically (if the parent had positive reward).

To select the final action to execute in the environment, the UCT agent selects a visited action with the maximum estimated value. We could consider alternate approaches here, such as selecting uniformly at random from unexplored actions if none of the visited actions have high enough expected

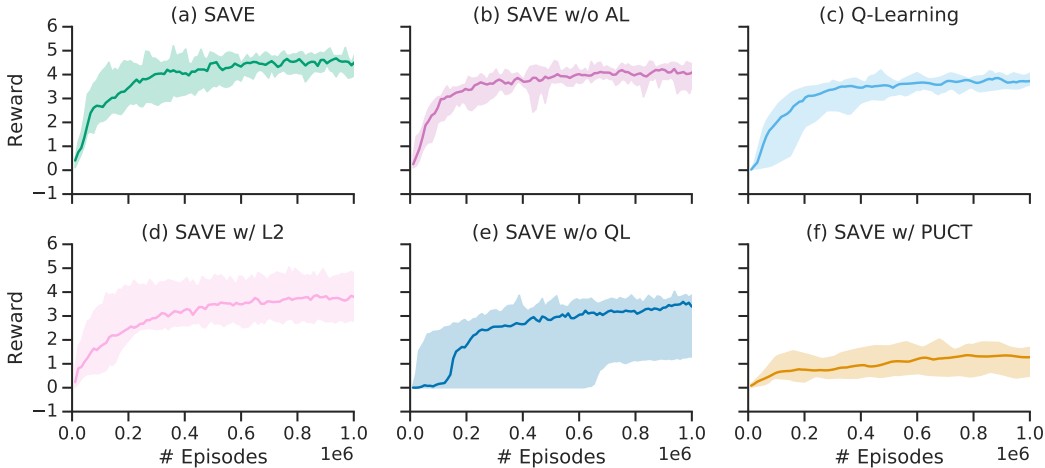

Figure C.1: Learning curves on the Covering task. Each plot shows median performance across 10 seeds, with shaded regions showing the min and max seed.

values. We experimented with this approach, using a threshold value of zero (which is the expected value for bad actions in Tightrope), and find that this indeed improves performance ($p = 0.02$), though the effect size is quite small: on the dense setting with $M = 95\%$ we achieve a median reward of 0.08 (using this thresholding action selection policy) versus 0.07 (selecting the max of visited actions).

## B.3 FUNCTION APPROXIMATION EXPERIMENTS

We used the same learning setup for the Q-learning, SAVE, and PUCT agents as described in Appendix A. For the network architecture of our agents, we used a shared multilayer perceptron (MLP) torso with two layers of size 64 and ReLU activations. To predict Q-values, we used an MLP head with two layers of size 64 and ReLU activations, with a final layer of size 100 (the number of actions) with a linear activation. To predict a policy in the PUCT agent, we used the same network architecture as the Q-value head. To predict state values in the PUCT agent, we used a separate MLP head with two layers of size 64 and ReLU activations, and a final layer of size 1 with a linear activation. All network weights were initialized using the default weight initialization scheme in Sonnet (Reynolds et al., 2017). For both the SAVE and PUCT agents we used loss coefficients of $\beta_Q = 0.5$ and $\beta_A = 0.5$.

We trained each agent 10 times and report results after 1e6 episodes in a version of Tightrope that has 95% terminal actions Figure 2, right). During training, the SAVE and PUCT agents had access to a search budget of 10 simulations; the Q-learning agent did not use search. We also explored training agents with different numbers of terminal actions and different budgets. Qualitatively, we found the same results as in the tabular setting: the PUCT agent can perform well for larger budgets (50+), but struggles with small budgets, underperforming the model-free Q-learning agent. In contrast, SAVE performed well in all our experiments, even for small budgets like 5 or 10.

## C DETAILS ON CONSTRUCTION

### C.1 AGENT DETAILS

**SAVE** For SAVE, we annealed $\beta_Q$ from 1 to 0.1 and $\beta_{\mathrm{PI}}$ from 0 to 4.5 over the course of 5e4 episodes. We found this allowed the agent to rely more on Q-learning early on in training to build a good Q-value prior, and then more on MCTS later in training once a good prior had already been established.

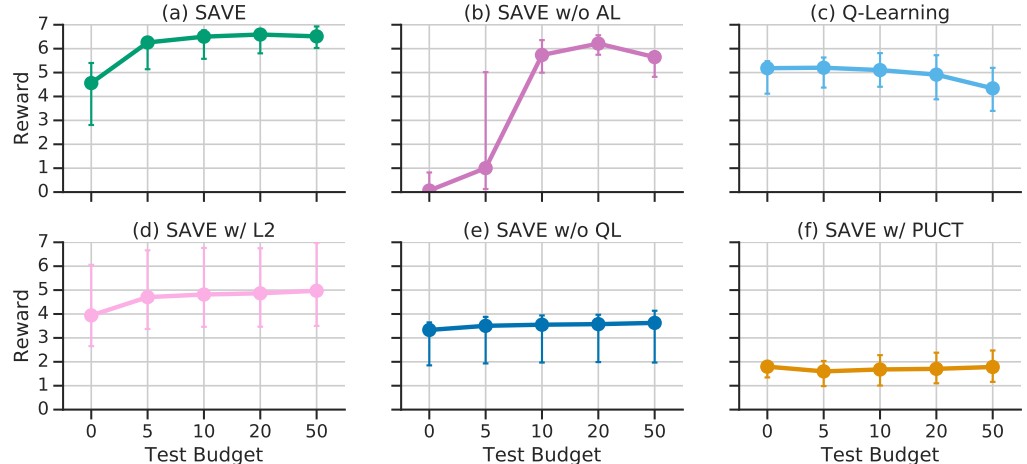

Figure C.2: Detailed final results on the Covering task. Each plot shows median performance across 10 seeds, with error bars showing the min and max seed.

**Q-Learning** The Q-Learning agent is as described in Section A.1. In particular, we follow the same setup as the GN-DQN agent described in Bapst et al. (2019). During training, we use pure Q-learning with no search. At test time, we may allow the Q-learning agent to additionally perform MCTS, using the same search procedure as that used by SAVE (i.e., initializing the Q-values using the trained Q-function and initializing the visit counts to one).

**SAVE without Amortization Loss** The SAVE without an amortization loss is the same as the basic SAVE agent, except that it includes no amortization loss (i.e., $\mathcal{L}(\theta, \mathcal{D}) = \beta_Q \mathcal{L}_Q(\theta, \mathcal{D})$). This is equivalent to the GN-DQN-MCTS agent described by Bapst et al. (2019).

**UCT** UCT is as described in Section 3.1, with $V(s)$ at unexplored nodes estimated via using a pretrained action-value function (trained using the same setup as the Q-learning agent). Additionally, unlike standard UCT we did not require all actions to be visited before descending down the search tree.

**SAVE with L2** SAVE with an L2 loss is identical to SAVE except that it uses a different amortizaton loss:

$$\mathcal{L}_A(\theta, \mathcal{D}) = \frac{1}{N} \sum_D \left\| Q_{\mathrm{MCTS}}(s, \cdot) - Q_\theta(s, \cdot) \right\|_2$$

Similar to the SAVE agent, we anneal $\beta_Q$ from 1 to 0.1 and $\beta_A$ from 0 to 0.045 over the course of 5e4 episodes.

**SAVE without Q-Learning** SAVE without the Q-learning loss is identical to SAVE except that we do not use Q-learning and we use the L2 amortization loss described in the previous paragraph:

$$\mathcal{L}(\theta, \mathcal{D}) = \beta_A \mathcal{L}_A(\theta, \mathcal{D})$$

where we set $\beta_A = 0.025$. The reason we use the L2 loss rather than the cross-entropy loss is that otherwise the Q-values will not actually be real Q-values, in that they will not have grounding in the actual scale of rewards. We did experiment with using only the cross-entropy loss with no Q-learning, and found slightly worse performance than when using the L2 loss and no Q-learning.

**SAVE with PUCT** SAVE with PUCT uses the same learning procedure as SAVE but a different search policy. Specifically, we use the PUCT search policy described in Section A.3 and Equation 7. To do this, we set $\pi(s, a) = \sigma(Q_\theta(s, a))$, where $\sigma$ is the softmax over actions with a temperature of 1. We use the same settings for Dirchlet noise to encourage exploration during search. After search is complete, we select an action using the same epsilon-greedy action procedure used by the SAVE

agent rather than selecting based on visit counts. We experimented with selecting based on visit counts instead, but found this resulted in the same level of performance.

## C.2 EXPERIMENTAL SETUP

Observations are given as graphs representing the scene, with objects in the scene corresponding to nodes in the graph and edges between every pair of objects. All agents use the same network architecture (Battaglia et al., 2018) described in Bapst et al. (2019) to process these graphs. Briefly, we use a graph network architecture which takes a graph as input and returns a graph with Q-values on the edges of the graph. Each edge corresponds to a relative object-based action like "pick up block B and put it on block D". Each edge additionally has multiple actions associated with it which correspond to particular offset locations where the block should be placed, such as "on the top left".

Bapst et al. (2019) describe four Construction tasks: Silhouette, Connecting, Covering, and Covering Hard. We reported results on three of these tasks in the main text (Connecting, Covering, and Covering Hard). The agents in Bapst et al. (2019) already reached ceiling performance on Silhouette and thus we do not report results for that task here, except to report that SAVE also reaches ceiling performance.

The agents used 10 MCTS simulations during training and were evaluated on 0 to 50 simulations at test time, with the exception of the UCT agent, which always used 1000 simulations at test time, and the Q-learning agent, which did not peform search during learning. We trained 10 seeds per agent and report results after 1e6 episodes. Figure C.1 show details of learning progress for each of the agents compared in the ablation experiments on the Covering task (Section 4.2), and Figure C.2 shows detailed final performances evaluated at different test budgets. We evaluated all agents on the hardest level of difficulty of the particular task they were trained on for either 10000 episodes (Figure 3a-c) or 1000 episodes (Figure 3d and Figure C.2). In general, while we find that search at test time can provide small boosts in performance, the main gains are achieved by incorporating search during training.

## C.3 DISCUSSION OF ABLATION RESULTS

Here we expand on the results presented in the main text and in Figure 3d and Figure C.2.

**Cross-entropy vs. L2 loss**    While the L2 loss (Figure C.2, orange) *can* result in equivalent performance as the cross-entropy loss (Figure C.2, green), this is at the cost of higher variance across seeds and lower performance on average. This is likely because the L2 loss encourages the Q-function to exactly match the Q-values estimated by search. However, with a search budget of 10, those Q-values will be very noisy. In contrast, the cross-entropy loss only encourages the Q-function to match the overall distribution shape of the Q-values estimated by search. This is a less strong constraint that allows the information acquired during search to be exploited while not relying on it too strongly. Indeed, we can observe that the agent with L2 amortization loss actually performs worse than the agent that has no amortization loss at all (Figure C.2, purple) when using a search budget of 10, suggesting that trying to match the Q-values during search too closely can harm performance.

Additionally, we can consider an interesting interaction between Q-learning and the amortization loss. Due to the search locally avoiding poor actions, Q-learning will rarely actually operate on low-valued actions, meaning most of its computation is spent refining the estimates for high-valued actions. The softmax cross entropy loss ensures that low-valued actions have lower values than high-valued actions, but does not force these values to be exact. Thus, in this regime we should have good estimates of value for high-valued actions and worse estimates of value for low-valued actions. In contrast, an L2 loss would require the values to be exact for *both* low and high valued actions. By using cross entropy instead, we can allow the neural network to spend more of its capacity representing the high-valued actions and less capacity representing the low-valued actions, which we care less about in the first place anyway.

**With vs. without Q-learning**    Without Q-learning (Figure C.2, teal), the SAVE agent's performance suffers dramatically. As discussed in the previous section, the Q-values estimated during search are very noisy, meaning it is not necessarily a good idea to try to match them exactly. Additionally, $Q_{\mathrm{MCTS}}$ is on-policy experience and can become stale if $Q_\theta$ changes too much between

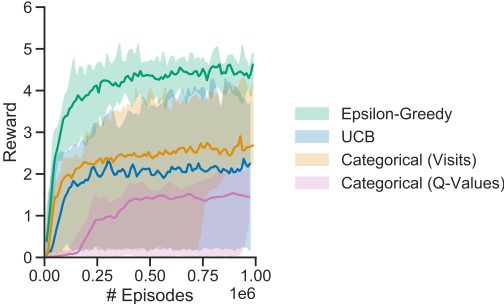

Figure C.3: Performance of different exploration strategies on the Covering task.

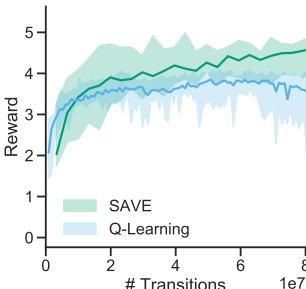

Figure C.4: Performance of SAVE and Q-learning on Covering, controlling for the same number of environment interactions (including those seen during search).

when $Q_{\mathrm{MCTS}}$ was computed and when it is used for learning. Thus, removing the Q-learning loss makes the learning algorithm much more on-policy and therefore susceptible to the issues that come with on-policy training. Indeed, without the Q-learning loss, we can *only* rely on the Q-values estimated during search, resulting in much worse performance than when Q-learning is used.

**UCT vs. PUCT**    Finally, we compared to a variant which utilizes prior knowledge by transforming the Q-values into a policy via a softmax and then using this policy as a prior with PUCT, rather than using it to initialize the Q-values (Figure C.2, brown). With large amounts of search, the initial setting of the Q-values should not matter much, but in the case of small search budgets (as seen here), the estimated Q-values do not change much from their initial values. Thus, if the initial values are zero, then the final values will also be close to zero, which later results in the Q-function being regressed towards a nearly uniform distribution of value. By initializing the Q-values with the Q-function, the values that are regressed towards may be similar to the original Q-function but will not be uniform. Thus, we can more effectively reuse knowledge across multiple searches by initializing the Q-values with UCT rather than incorporating prior knowledge via PUCT.

## C.4    ADDITIONAL RESULTS

We performed several other experiments to tease apart the questions regarding exploration strategy and data efficiency.

**Exploration strategy**    When selecting the final action to perform in the environment, SAVE uses an epsilon-greedy exploration strategy. However, many other exploration strategies might be considered, such as UCB, categorical sampling from the softmax of estimated Q-values, or categorical sampling from the normalized visit counts. We evaluated how well each of these exploration strategies work, with the results shown in Figure C.3. We find that using epsilon-greedy works the best out of these exploration strategies by a substantial margin. We speculate that this may be because it is important for the Q-function to be well approximated across all actions, so that it is useful during MCTS backups. However, UCB and categorical methods will not uniformly sample the ac-

tion space, meaning that some actions are very unlikely to be ever learned from. The amortization loss will not help either, as these actions will not be explored during search either. The error in the Q-values for unexplored actions will grow over time (due to catastrophic forgetting), leading to a poorly approximated Q-function that is unreliable. In contrast, epsilon-greedy consistently spends a little bit of time exploring these actions, preventing their values from becoming too inaccurate. We expect this would be less of a problem if we were to use a separate state-value function for bootstrapping (as is done by AlphaZero).

**Data efficiency** With a search budget of 10, SAVE effectively sees 10 times as many transitions as a model-free agent trained on the same number of environment interactions. To more carefully compare the data efficiency of SAVE, we compared its performance to that of the Q-learning agent on the Covering task, controlling for the same number of environment interactions (including those seen during search). The results are shown in Figure C.4, illustrating that SAVE converges to higher rewards given the same amount of data. We find similar results in the Marble Run environment, shown in Figure D.2.

## D  DETAILS ON MARBLE RUN

### D.1  SCENE GENERATION

Scenes contain the following types of objects (similar to Bapst et al. (2019)):

- Floor (in black) that supports the blocks placed by the agent.
- Available blocks (row of blue blocks at the bottom) that the agent picks and place in the scene (with replacement).
- Blocks (blue blocks above the floor) that the agent has already placed. They may take a lighter blue color to indicate that they are *sticky*. A *sticky* block gets glued to anything it touches.
- Goal (blue dot) that the agent has to reach with the marble.
- Marble (green circle) that the agent has to route to the goal.
- Obstacles (red blocks, including two vertical walls), that the agent has to avoid, by not touching them neither with the blocks or the marble.

All the initial positions for obstacles in the scene are sampled from a tessellation (similar to the *Silhouette* task in Bapst et al. (2019)) made of rows with random sequences of blocks with sizes of 1 discretization unit in height and 1 or 2 discretization units in width (a discretization unit corresponds to the side of the first available block). The sampling process goes as follows:

1. Set the vertical position of the goal to the specified discrete height (according to level) corresponding to the center of one of the tessellation rows, and the vertical position of the marble 2 rows above that.

2. Uniformly sample a horizontal distance between the marble and the goal from a predefined range, and uniformly sample the absolute horizontal positions respecting that absolute distance.

3. Sample a number of obstacles (according to level) from the tessellation spanning up to the vertical position of the marble.

Obstacles are sampled from the tessellation sequentially. Before each obstacle is sampled, all objects in the tessellation that are too close ($\pm$ 2 layers vertically and with less than 2 discretization units of clearance sideways) to the goal, the target, or previously placed obstacles, are removed from the tessellation in order to prevent unsolvable scenes. Then probabilities are assigned to all of the remaining objects in the tessellation according to one of the following criteria (the criteria itself is also picked randomly with different weights) designed to avoid generating trivial scenes:

- (Weight=4) Pick uniformly a tessellation object lying exactly on the floor and between the marble and the goal horizontally, since those objects prevent the marble from rolling freely on the floor (only applicable if the tessellation still has objects of this kind available).

- (Weight=1) Pick a tessellation object that is close (horizontally) to the marble. Probabilities proportional to $\frac{1}{(d/\tau)^2+0.1}$ (where $d$ is the horizontal distance between each object and the marble scaled by the width of the scene and $\tau$ is a temperature set to 0.1) are assigned to all objects left in the tessellation, and one of them is picked.

- (Weight=1) Pick a tessellation object that is close (horizontally) to the goal. Identical to the previous one, but using the distance to the goal.

- (Weight=1) Pick a tessellation object that is close (horizontally) to the middle point between the ball and the goal. Identical to the previous one, but using the distance to the middle point, and a temperature of 0.2.

- (Weight=1) Pick any object remaining in the tessellation with uniform probability (to increase diversity).

## D.2 CURRICULUM DIFFICULTY

We used a curriculum to sample scenes of increasing difficulty (Fig. D.1) according to:

| Level | Goal height (discretization units) | Marble/Goal distance (scene width fraction) | # obstacles | Max # steps |
|---|---|---|---|---|
| 0 | 0 | [0.03,0.3] | 1 | 20 |
| 1 | 0 | [0.36,0.49] | 1 | 20 |
| 2 | 0 | [0.50,0.63] | 2 | 20 |
| 3 | 0 | [0.69,0.82] | 2 | 20 |
| 4 | 0 | [0.83,1] | 3 | 20 |
| 5 | 1 | [0.83,1] | 3 | 25 |
| 6 | 2 | [0.83,1] | 4 | 30 |

During both training and testing, episodes at a certain curriculum level are sampled not only from that difficulty, but also from all of the previous difficulty levels, using a truncated geometric distribution with a decay of 0.5. This means that at each level, about half of the episodes correspond to that level, half of the remaining episodes correspond to the previous level, half of the remaining to the level before that, and so on. By truncated we mean that, because it is not possible to sample episodes for negative levels, so we truncate the probabilities there and re-normalize.

## D.3 ADAPTIVE CURRICULUM

Given the complexity and the sparsity of rewards in this task, we trained agents using an adaptive curriculum to avoid presenting unnecessarily hard levels to the agent until the agent is able to solve the simpler levels. Specifically at each level of the curriculum we keep track and bin past episode results according to all possible combinations of scene properties consisting of:

- Height of the target (discretized to tessellation rows).

- Horizontal distance $d$ between marble and goal (discretized to $d < 1/3$, $1/3 < d < 2/3$, or $d > 2/3$, where d is normalized by the width of the scene).

- Number of obstacles.

- Height of the highest obstacle (discretized to tessellation rows).

- Height of the lowest obstacle (discretized to tessellation rows).

and require the agents to have solved at least 50% of scenes of the last 50 episodes in each bin individually, but simultaneously in all bins[3]. before we allow the agent to progress to the next level of difficulty. This is a very strict criteria, which effectively means the agent has to find solutions for all representative variations of the task at that level before is allowed to progress to the next level.

---

[3]Ignoring underrepresented bins for which episodes are generated at less than 0.25 the expected generation rate according to a uniform prior across bins (which we only start estimating after the first 300 episodes, to discover a sufficient number of distinct bins). This is to avoid delayed curriculum progress just due to lack of recent statistics for those bins.

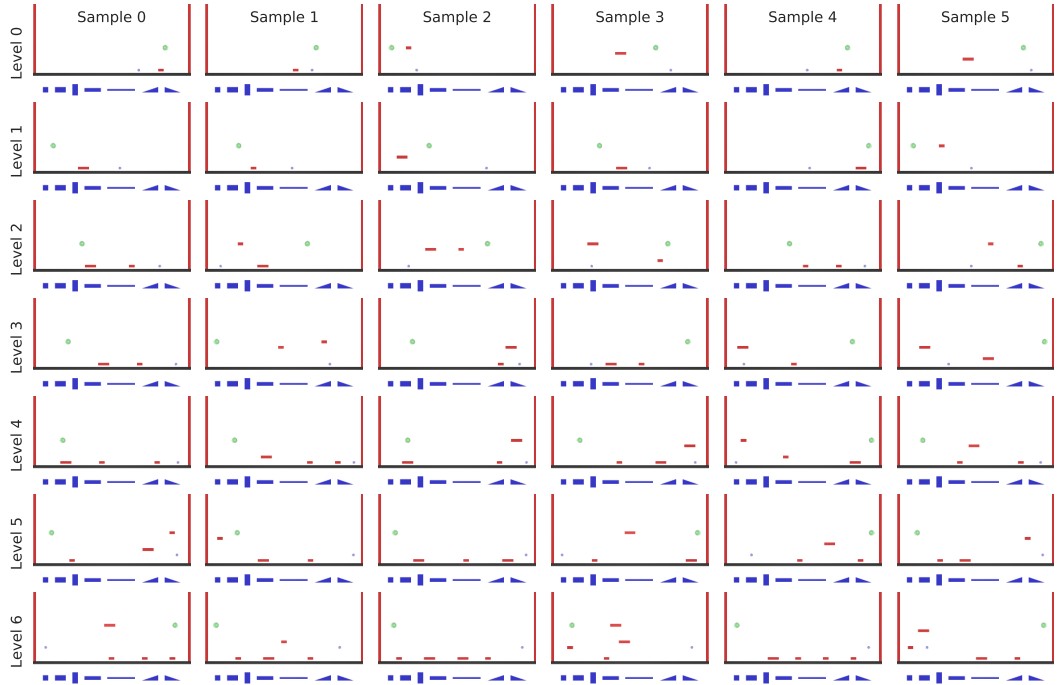

Figure D.1: Scenes samples at each curriculum level for the marble run task. During training, the $n$-th level of the curriculum consists of scenes sampled from the rows up to the $n$-th row with a truncated geometric distribution with a decay of 0.5.

## D.4 AGENT STEP, ACTION AND REWARD EVALUATION

Each agent step consists of four phases:

1. Block placement phase: The agent picks one object from the available objects and places it into the scene. If the block placed by the agent was *sticky* the agent will receive a negative reward according to the cost (which may be either 0 or 0.04).

2. Block settlement phase: The physics simulation (keeping the marble frozen) is run until the placed blocks settle (up to a maximum of 20 s). During this phase the new block may affect the position of previously placed blocks.

3. Marble dynamics phase: The physics simulation including the marble is run until the marble collides with 8 objects, with a timeout of 10 s at each collision, that is a maximum of 80s. This phase may terminate early if the marble reaches the goal (task is solved and episode terminated with a reward of 1.), but also if the marble or any of the blocks touch an obstacle.

4. Restore state phase: After the marble dynamics phase, the marble and all of the blocks are moved back to the position where they were at the end of the block settlement phase. This is to prevent the agent from using the marble to indirectly move the blocks with a persistent effect across steps.

The block placement phase and block settlement phase, as well as the action space is identical to those in Bapst et al. (2019).

## D.5 OBSERVATION

The observation is identical to the Construction tasks in Bapst et al. (2019), with an additional one-hot encoding of the object shape (e.g. rectangle vs triangle vs circle) and includes all blocks positions and the initial marble position at the end of the block settlement phase. Note that the agent

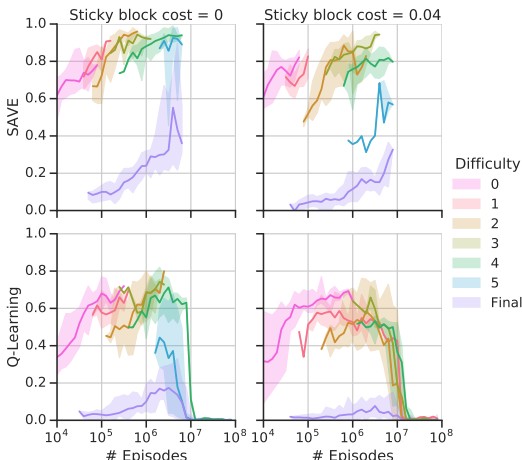

Figure D.2: Learning curves for the Marble Run environment. Each line shows the median reward across 10 seeds, and the shaded regions show min and max seed performance. Each color corresponds to a different level of curriculum difficulty. Difficulties less than the final difficulty are only evaluated while the agent is training at that curriculum level; the final level of difficulty is always evaluated.

never actually gets to observe the marble's dynamics, and therefore does not get direct feedback about *why* the marble does or does not make it to the goal (such that it is getting stuck in a hole). An interesting direction for future work would be to incorporate this information into the agent's learning as well.

## D.6 TERMINATION CONDITION

There are several episode termination conditions that may be triggered before the task is solved:

- An agent places a block in a position that overlaps with an existing block or obstacle.
- An agent has placed a block that during the settlement phase touches an obstacle.
- An agent has placed a block that, at the end of the block settlement phase overlaps with the initial marble position.
- Maximum number of steps is reached.

Note that touching obstacles during the marble dynamics phase does not terminate the episode because we are purely evaluating the reward function and, during the restore state phase, all objects are returned to there previous locations. This makes it possible for the agent to correct for any obstacle collisions that happened during the marble dynamics phase, by placing additional blocks that re-route the marble.

## D.7 ADDITIONAL RESULTS

We used the same experimental setup as in the other Construction tasks (Appendix C). In particular, during training, for each seed of each agent we checkpoint the weights which achieve the highest reward on the highest curriculum level, and then use these checkpoints to evaluate performance in Figure 4. Figure D.2 additionally shows details of the training performance at each level of difficulty in the curriculum. We can see that at around difficulty level 4-5, the Q-learning agent becomes unstable and crashes, while the SAVE agent stays stable and continues to improve. Indeed, as shown in Figure D.3, the Q-learning agent never makes it to difficulty level 6 (when sticky blocks are free) or even difficulty level 5 (when sticky blocks have a moderate cost). The SAVE agent is able to reach harder levels of difficulty, and does so with fewer learning steps.

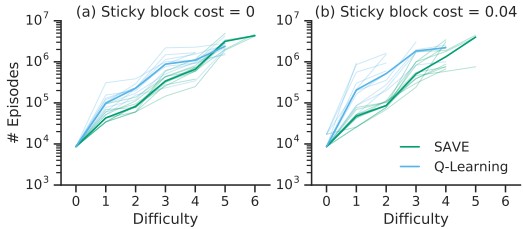

Figure D.3: Curriculum progress in Marble Run. Light lines show individual curriculum progress per seed, and dark lines are computed over the median of these seeds. The $x$-axis shows the particular curriculum level and the $y$-axis indicates at which episode that level of difficulty was reached.

| Level | Baseline | Controlled | SAVE | % Change |
|-------|----------|-----------|------|----------|
| Alien | 71925.1 | 96013.5 | **280227.3** | 191.9% |
| Asteroids | 251033.3 | **266306.7** | **274431.7** | 3.1% |
| Beam Rider | 96654.4 | 113930.6 | **195703.8** | 71.8% |
| Centipede | 517332.2 | 562742.3 | **767206.6** | 36.3% |
| Crazy Climber | **311203.8** | 271151.5 | **324726.4** | 19.8% |
| Frostbite | 15814.2 | 11052.3 | **202744.2** | 1734.4% |
| Gravitar | 7854.0 | **11314.3** | **11484.1** | 1.5% |
| Hero | 30515.9 | **44574.3** | **44796.0** | 0.5% |
| Ms. Pacman | 25377.4 | 27776.3 | **47186.0** | 69.9% |
| Name This Game | 45027.1 | 40790.0 | **58621.1** | 43.7% |
| River Raid | 33819.5 | 32720.8 | **41031.6** | 25.4% |
| Space Invaders | 3639.2 | 42387.4 | **63684.7** | 50.2% |
| Up 'n' Down | **563661.0** | **568735.6** | **585475.6** | 2.9% |
| Zaxxon | 116892.6 | 73073.1 | **213370.4** | 192.0% |
| Median | 58476.1 | 58823.7 | 199224.0 | 40.0% |
| Mean | 149339.3 | 154469.2 | 222192.1 | 174.5% |

Table E.1: Results on Atari. Scores are final performance averaged over 3 seeds. "Baseline" is the standard version of R2D2 (Kapturowski et al., 2018). "Controlled" is our version that is controlled to have the same replay ratio as SAVE. The rightmost column reports the percent change in reward of SAVE over the controlled version of R2D2. Bold scores indicate scores that are within 5% of the best score on a particular game. The last two rows show median and mean scores, respectively. The percentages in the last two rows show the median and mean across percent change, rather than the percent change of the median/mean scores.

# E DETAILS ON ATARI

## E.1 EXPERIMENTAL SETUP

We evaluated SAVE on a set of 14 Atari games in the Arcade Learning Environment (Bellemare et al., 2013). The games were chosen as a combination of classical action Atari games such as *Asteroids* and *Space Invaders*, and games with a stronger strategic component such as *Ms. Pacman* and *Frostbite*, which are commonly used as evaluation environments for model-based agents (Buesing et al., 2018; Farquhar et al., 2018; Oh et al., 2017; Guez et al., 2019).

SAVE was implemented on top of the R2D2 agent (Kapturowski et al., 2018) as described in Algorithm A.1. Concretely, this means we evaluate the function $Q_{\mathrm{MCTS}}$ instead of $Q_\theta$ to select an action in the actors, and optimize the combined loss function (Equation 6) instead of the TD loss in the learner. For hyperparameters, we used a search budget of 10, and $\beta_Q = 1$, $\beta_A = 10$. We did very little tuning to select these hyperparameters, only sweeping over two values of $\beta_A \in \{1, 10\}$. We found while both of these settings resulted in similar performance, $\beta_A = 10$ worked slightly better. It is likely that with further tuning of these parameters, even larger increases in reward be achieved,

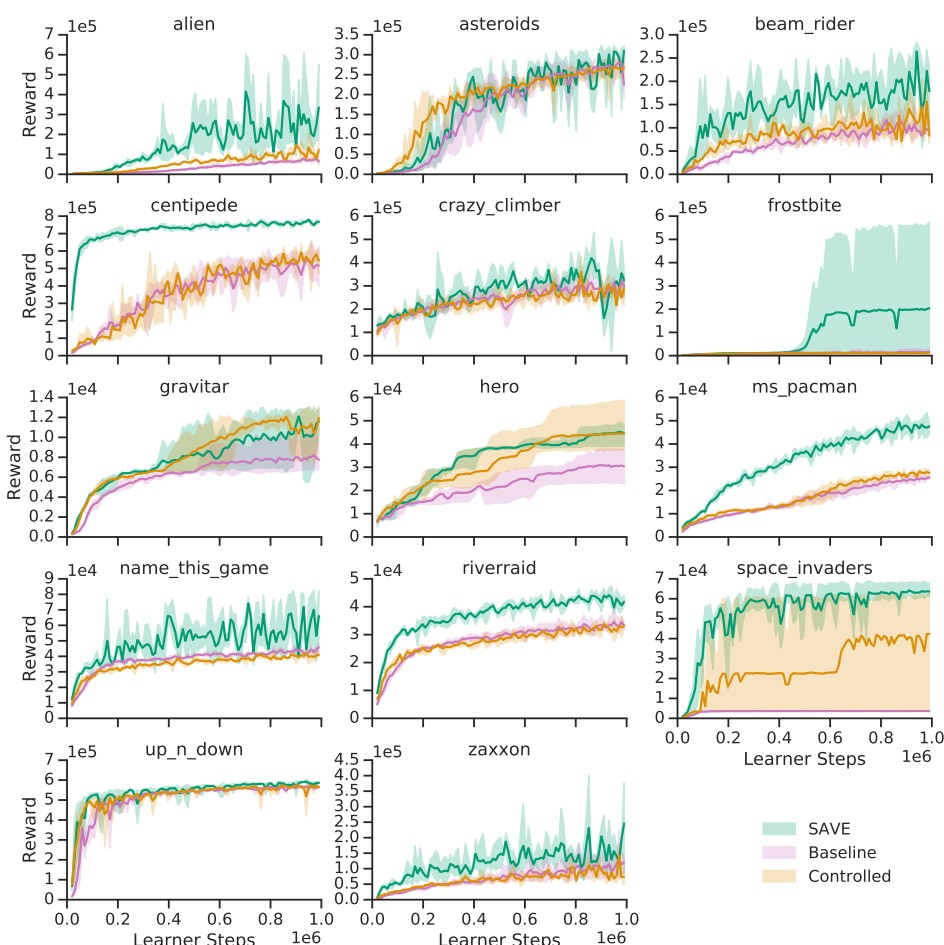

Figure E.1: Learning curves on Atari games. Solid lines show the average over 3 seeds, and shaded regions show min and max seeds.

as $\mathcal{L}_Q$ and $\mathcal{L}_A$ will have very different relative magnitudes depending on the scale of the rewards in each game.

All hyper-parameters of R2D2 remain unchanged from the original paper, with the exception of actor speed compensation. By running MCTS, multiple environment interactions need to be evaluated for each actor step, which means transition tuples are added to the replay buffer at a slower rate, changing the replay ratio. To account for this, we increase the number of actors from 256 to 1024, and change the actor parameter update interval from 400 to 40 steps.

### E.2 EVALUATION

The learning curves of our experiment are shown in Figure E.1, and Table E.1 shows the final performance in tabular form. We ran three seeds for each of the Baseline, Controlled and SAVE agents for each game and computed final scores as the average score over the last 2e4 episodes of training. The Baseline agent represents the unchanged R2D2 agent from (Kapturowski et al., 2018). The Controlled agent is a R2D2 agent controlled to have the same replay ratio as SAVE, which we achieve by running MCTS in the actors but then discarding the results. As in SAVE, we use 1024 actors with update interval 40 for the controlled agent.

We can observe that in the majority of games, SAVE performs not only better than the controlled agent but also better than the original R2D2 baseline. While we see big improvements in the strategic games such as Ms. Pacman, we also notice a gain in many of the action games. This suggests that

model-based methods like SAVE can be useful even in domains that do not require as much long-term reasoning.

