# OpenReview forum: "Combining Q-Learning and Search with Amortized Value Estimates"
_ICLR.cc/2020/Conference — Accept (Poster)_

### Official Review · AnonReviewer3 · 2019-10-25
**Official Blind Review #3**

**Rating:** 6

**Review:**

This paper proposes SAVE that combines Q learning with MCTS. In particular, the estimated Q values are used as a prior in the selection and backup phase of MCTS, while the Q values estimated during MCTS are later used, together with the real experience, to train the Q function. The authors made several modifications to ‘standard’ setting in both Q learning and MCTS. Experimental results are provided to show that SAVE outperforms generic UCT, PUCT, and Q learning.

Overall the paper is easy to follow. The idea of the paper is interesting in the sense that it tries to leverage the computation spent during search as much as possible to help the learning. I am not an expert in the of hybrid approach, so I can not make confident judgement on the novelty of the paper.

 The only concern I have is that the significance of the result in the paper:
1. The proposed method, including the modifications to MCTS and Q learning (section 3.2 and 3.3), is still a bit ad-hoc. The paper has not really justified why the proposed modification is a better choice except a final experimental result. Some hypotheses are made to explain the experimental results. But the authors have not verified those hypotheses. Just to list a few here: (a). The argument made in section 2.2 about count based prior; (b). the statement of noisy Q_MCTS to support the worse performance of L2 loss in section 4.2; (c). In the last paragraph of section 4.3, why would a model free agent with more episodes results in worse performance?
2. The baselines used in this paper are only PUCT and a generic Q learning. What are the performances of other methods that are mentioned in section 2.1, like Gu 2016, Azizzadenesheli 2018, Bapst 2019?

Other comments:
1. What is the performance of tabular Q-learning in Figure 2 (a-c)?


**Experience Assessment:**

I do not know much about this area.

**Review Assessment: Checking Correctness Of Derivations And Theory:**

I assessed the sensibility of the derivations and theory.

**Review Assessment: Checking Correctness Of Experiments:**

I assessed the sensibility of the experiments.

**Review Assessment: Thoroughness In Paper Reading:**

I read the paper at least twice and used my best judgement in assessing the paper.

---

> ### Author Response · Authors · 2019-11-07
> **Response to Reviewer #3**
>
> Thank you very much for your comments! We are glad to hear that you think the paper is clear and that the idea is interesting.
>
> 1. We respectfully disagree that SAVE is ad-hoc. Our primary baselines—SAVE w/o AL and PUCT—are methods that have been previously published, and both suffer from potential issues during training which SAVE explicitly addresses. We tried to emphasize this in the text, but perhaps the justification for SAVE’s approach was not sufficiently clear. We will attempt to clarify this in our revision, and provide further justification below:
>
> (a) In Section 2.2, we hypothesized that approaches which use a count-based policy prior will suffer in environments with high branching factors and small search budgets. We formulated the Tightrope domain as a way to test this hypothesis, and our experiments demonstrate that our hypothesis is correct. Since value is the quantity that we actually want to maximize, it makes more sense—and results in better performance—if we regress towards values rather than regressing towards visit counts.
>
> Similarly, we hypothesized in Section 2.1 that the reason prior approaches combining Q-learning and MCTS found unstable performance is due to the fact that the experience generated by MCTS is too off policy and does not allow the Q-function to learn about bad actions. Our experiments on the Construction tasks comparing against the SAVE w/o AL baseline similarly demonstrate that this hypothesis is correct: by including information about bad actions through the amortization loss, the Q-function becomes much more stable.
>
> (b) Regarding why the L2 loss works less well, we agree that our justification is post-hoc and requires further investigation. However, we believe the empirical performance justifies using the cross-entropy loss instead, and that a more detailed explanation of the difference is more a topic for future work.
>
> (c) SAVE achieves better performance than a model-free agent trained with more data because it allows the agent to try out multiple potential actions at each point in time. While the agent may estimate these actions to have similar values, search allows it to uncover imperfections in those estimates and avoid locally suboptimal actions (such as an action that causes the episode to terminate). In contrast, a model-free agent does not have the ability to try out and compare multiple actions. If it ends up taking an action that causes the episode to terminate, then it will have to restart from the beginning. Thus, SAVE allows the agent to gather experience from later in the episode than model-free agents, resulting in better performance even when controlling for the same number of environment transitions.
>
> 3. The SAVE w/o AL agent is based on the GN-DQN-MCTS agent described by Bapst et al. (2019) (as stated in the second paragraph of Section 4.2), and is similar in spirit to other approaches which include planning in the training loop but which do not leverage the value computations performed during search (Gu et al. 2016, Azizzadenesheli et al. 2018). The relevant aspect of these papers is this common property (that they do not use the computed values from search), and thus we feel that the SAVE w/o AL baseline is a sufficient comparison. Additionally, there are many other choices made by the other papers which would make them inappropriate to directly compare to. Specifically, Gu et al. were concerned on continuous Q-learning and evaluated on Mujoco continuous control tasks. Azizzadenesheli et al. were concerned with learning a model using GANs and using that within MCTS; while they evaluated their approach on Atari they did not achieve better performance than an agent trained with DDQN. In contrast, in our Atari experiments SAVE strongly outperforms our model-free baseline (R2D2).
>
> Other comments:
>
> 1. This is a good point; we should have included a comparison to tabular Q-learning as well. We are working on this and will post a later revision with these results.

---

> > ### Comment · AnonReviewer3 · 2019-11-15
> > **After Rebuttal**
> >
> > Thanks for the response. I don't think the rebuttal address all of my questions. My score will remain the same.
> >
> > To be more precise about the 'hypotheses' comment. I don't think a final performance score is convincing enough. For example, after hypothesizing the failure modes of the count based prior,  is it possible to show that it is indeed happening in the experiments? What is the action 'A' that is recommend by Q-learning but modified by planning, and thus not have been updated? I would also argue that the response to (c)  is again making hypotheses rather than evidence.

---

> > > ### Author Response · Authors · 2019-11-15
> > > **Response to reviewer**
> > >
> > > Thank you for your reply. We agree it is always possible to run more experiments to further tease apart the details of our findings, and we hope that future work will build upon this paper to do so.
> > >
> > > Our experiments were specifically designed to test the referenced hypotheses; specifically:
> > >
> > > In the comparison to PUCT, our hypothesis is that count-based priors should fail in regimes with small search budgets, large branching factors, and many bad actions, which is exactly what we find in Tightrope (which was precisely designed to have high branching factors with many bad actions). We also find these policies will spend most of their time re-expanding terminal actions, indicating that the prior has collapsed. We are happy to add this additional result to the paper.
> > >
> > > In the comparison to SAVE without AL, we hypothesized that actions which the underlying Q-function thinks are good are not actually executed (because search ends up avoiding them), thus leading to a poorly approximated Q-function. Our experiments confirm this, because they show that when SAVE w/o AL does not have access to search, its performance is extremely poor: in other words, the Q-function is indeed very bad. If it would help, we could provide some additional statistics such as the proportion of time the search causes the agent to take a different action, or how much the Q-values change as a result of search.
> > >
> > > We agree that our response in (c) is a post-hoc justification and that it warrants further investigation. However, we also see this as outside the scope of the present paper, which is to investigate the effect of different choices about how to use the knowledge gained from search (and not to explain why model-based methods can perform better than model-free).

---

### Official Review · AnonReviewer2 · 2019-10-25
**Official Blind Review #2**

**Rating:** 6

**Review:**

This paper proposes Search with Amortized Value Estimates (SAVE), which combines Q-learning and Monte-Carlo Tree Search (MCTS). SAVE makes use of the estimated Q-values obtained by MCTS at the root node (Q_MCTS), rather than using only the resulting action or counts to learn a policy. It trains the amortized value network Q_theta via the linear combination of Q-learning loss and the cross-entropy loss between the softmax(Q_MCTS) and softmax(Q_theta). Then, SAVE incorporates the learned Q-function into MCTS by using it for the initial estimate for Q at each node and for the leaf node evaluation by V(s) = max_a Q_theta(s,a). Experimental results show that SAVE outperforms the baseline algorithms when the search budget is limited.


- The idea of training Q-network using the result of MCTS planning is not new (e.g. UCTtoRegression in Guo et al 2014), but this paper takes further steps: the learned Q-network is again used for MCTS planning as Q initialization, the cross-entropy loss is used instead of L2-loss for the amortized value training, and the total loss combines Q-learning loss and the amortization loss.
- In Figure 1, it says that the final action is selected by epsilon-greedy. Since SAVE performs MCTS planning, UCB exploration seems to be a more natural choice than the epsilon-greedy exploration. Why does SAVE use a simple epsilon-greedy exploration? Did it perform better than UCB exploration or softmax exploration? Also, what if we do not perform exploration at all in the final action selection, i.e. just select argmax Q(s,a)? Since exploration is performed during planning, we may not need exploration for the final action selection?
- Can SAVE be extended to MCTS for continuous action space? SAVE trains Q-network, rather than a policy network that can sample actions, thus it seems to be more difficult to deal with continuous action space.
- In Eq. (5), we may introduce a temperature parameter that trade-offs the stochasticity of the policy to further improve the performance of SAVE.
- In Tightrope domain (sec 4.1), it says: "The MDP is exactly the same across episodes, with the same actions always having the same behavior.", but it also says: "In the sparse reward setting, we randomly selected one state in the chain to be the “final” state to form a curriculum over the length of the chain." It seems that those two sentences are contradictive.
- In the Tightrope experiment's tabular results (Figure 2), the performance of Q-learning is not reported. I want to see the performance of Q-learning here too.
- In Figure 2, the search budgets for training and testing are equal, which seems to be designed to benefit SAVE than PUCT. Why the search budget should be very small even during training? Even if the fixed and relatively large search budget (e.g. 50 or 100) is used during training and the various small search budgets are only used in the test phase, does SAVE still outperform PUCT?
- In Figure 2 (d), model-free Q-learning does not perform any planning, thus there will be much less interaction with the environment compared to SAVE or PUCT. Therefore, for a fair comparison, it seems that the x-axis in Figure 4-(d) should be the number of interactions with the environment (i.e. # queries to the simulator), rather than # Episodes. In this case, it seems that Q-Learning might be much more sample efficient than SAVE.
- In Figure 3, what is the meaning of the test budget for Q-Learning since Q-Learning does not have planning ability? If this denotes that  Q-network trained by Q-learning loss is used for MCTS, what is the difference between Q-Learning and SAVE w/o AL?
- In Figures 3, 4, 5, it seems that comparisons with PUCT are missing. In order to highlight the benefits of SAVE for efficient MCTS planning, the comparison with other strong MCTS baselines (e.g. PUCT that uses learned policy prior) should be necessary. A comparison only with a model-free baseline would not be sufficient.


-----
after rebuttal:

Thank the authors for clarifying my questions and concerns. I feel satisfied with the rebuttal and raise my score accordingly.

**Experience Assessment:**

I have published one or two papers in this area.

**Review Assessment: Checking Correctness Of Derivations And Theory:**

N/A

**Review Assessment: Checking Correctness Of Experiments:**

I carefully checked the experiments.

**Review Assessment: Thoroughness In Paper Reading:**

I read the paper at least twice and used my best judgement in assessing the paper.

---

> ### Author Response · Authors · 2019-11-07
> **Response to Reviewer #2 (1/2)**
>
> Thank you for your insightful comments and suggestions for future work! We have addressed these in detail below. However, we are unsure which of your comments were most important in deciding your score. Would you be able to clarify this?
>
> 1. Of course, we agree that the combination of Q-learning with MCTS is not new. We attempted to convey this in Sections 2.1 and 2.2, though perhaps we did not make it clear enough the difference between past work and our contributions. We will work on updating the language in the paper to be clearer on this. In particular, we emphasize that in contrast to previous approaches, SAVE is the first to simultaneously use MCTS to strengthen the Q-function, and the Q-function to strengthen MCTS. Moreover, SAVE addresses two important limitations of these previous approaches: that without a cross-entropy amortization loss, the Q-function will be poorly approximated; and that using the visit counts from search to improve the policy can be unreliable in the regime of small search budgets. We also find that the cross-entropy loss is quite crucial in our experiments to achieve good performance, compared to the L2 loss.
>
> Thank you for pointing out Guo et al. (2014); we were missing this reference and will add it to the paper.
>
> 2. We chose epsilon-greedy exploration because it is the standard choice of exploration for DQN agents, and it allowed for a more controlled comparison between SAVE and the model-free Q-learning. However, we agree it would be interesting to try using UCB exploration to select the final action. We will perform some experiments with this and update the paper with the results. We did try softmax exploration in the past but did not find it made a difference.
>
> 3. Extending SAVE to continuous action spaces is an important direction, but out of scope here (though we’re exploring it now). However, we emphasize that discrete problems constitute a large proportion of domains in deep RL, ranging from games (Atari, Go, etc.) to real-world applications like combinatorial optimization. Thus, exploring ways of improving discrete search is a valuable research direction in its own right.
>
> 4. Thank you for this suggestion! Although we have not experimented with a temperature parameter in the softmax functions we expect that better performance might be attained by tuning such a parameter. However, we leave this as an interesting direction for future work and will leave a comment about this in the paper.
>
> 5. Thank you for pointing out this out, we will clarify this in the text. Specifically, the behavior of the transition function is identical across episodes, with the exception of the behavior at the final state in the sparse reward setting.
>
> 6. This is a good point; we should have included a comparison to tabular Q-learning as well. We are working on this and will post a later revision with these results.
>
> 7. In many domains, a simulator may be available but also may be very slow. In our experiments, the simulator for the Construction tasks is a good example of this: training the agents with a search budget of 50 or 100 simulations is prohibitively costly. In real world environments, many simulators are extremely costly to run (such as physical simulators for fluid dynamics). Thus, it is an important area of research to demonstrate how such simulators can be effectively used even when we can rely on only very few simulations (both at training and at test time).
>
> As to how SAVE compares to PUCT with larger search budgets during training, we can see in Figure 2 that as the search budget increases, both methods converge reliably to the solution. If one has access to a fast simulator and can perform a significant amount of search, we see no reason not to do this. Rather, we are interested in scenarios in which a large amount of search is impractical to use and where PUCT-like methods will not perform well. We will update the text to emphasize this further.
>
> 8. We discuss this point at the end of Section 4.3. Additionally, please see our response to Reviewer #4 (comment 7).
>
> 9. We agree this is confusing and will update both the main text and the appendix to be clearer. Specifically, the “Q-learning” agent is an agent which uses regular Q-learning during training, and at test time additionally performs some amount of search (using the same version of MCTS as that used by SAVE). It is different from SAVE w/o AL in that the Q-learning agent may only perform search at test time, while SAVE w/o AL performs search both during training and testing.

---

> ### Author Response · Authors · 2019-11-07
> **Response to Reviewer #2 (2/2)**
>
> 10. The best performance we could achieve with PUCT on the Construction tasks is in the form of the SAVE w/ PUCT baseline in Figure 3. However, this agent still performs Q-learning and transforms the Q-values into a policy for use by the PUCT exploration term. If we train an actual policy prior and regress towards the visit counts, we find that performance is around zero in the Covering task (this result is actually what motivated our experiments in the Tightrope domain). This is because the Construction tasks have enormous branching factors (1000s-10000s of actions per state), and thus with a small search budget PUCT is unable to learn a useful policy for the reasons described in Section 2.2. Thus, we would not expect PUCT to work in the Marble Run task either. We will add further discussion of these results in the text.
>
> It is likely that PUCT would work better on Atari which has a small branching factor; however, the point of our Atari experiments was to show that SAVE can be dropped into existing Q-learning agents and achieve good performance with minimal effort. In contrast, even if we included a PUCT baseline, it would require significant work to tune the agent on Atari. We thus take this as an illustration of SAVE’s ease-of-use.

---

### Official Review · AnonReviewer4 · 2019-10-31
**Official Blind Review #4**

**Rating:** 6

**Review:**

This paper proposes an approach, named SAVE, which combines model-free RL (e.g. Q-learning) with model-based search (e.g. MCTS). SAVE includes the value estimates obtained for all actions available in the root node in MCTS in the loss function that is used to train a value function. This is in contrast to closely-related approaches like Expert Iteration (as in AlphaZero etc.), which use the visit counts at the root node as a training signal, but discard the value estimates resulting from the search.

The paper provides intuitive explanations for two situations in which training signals based on visit counts, and discarding value estimates from search, may be expected to perform poorly in comparison to the new SAVE approach:
1) If a trained Q-function incorrectly recommends an action "A", but a search process subsequently corrects for this and deviates from "A", no experience for "A" will be generated, and the incorrect trained estimates of this action "A" will not be corrected.
2) In scenarios with extremely low search budgets and extremely high numbers of poor actions, a search algorithm may be unable to assign any of the visit count budget to high-quality actions, and then only continue recommending the poor actions that (by chance) happened to get visits assigned to them.

The paper empirically compares the performance of SAVE to that of Q-Learning, UCT, and PUCT (the approach used by AlphaZero), on a variety of environments. This includes some environments specifically constructed to test for the situations described above (with high numbers of poor actions and low search budgets), as well as standard environments (like some Atari games). These experiments demonstrate superior performance for SAVE, in particular in the case of extremely low search budgets.

I would qualify SAVE as a relatively simple (which is good), incremental but convincing improvement over the state of the art -- at least in the case of situations with extremely low search budgets. I am not sure what to expect of its performance, relative to PUCT-like approaches, when the search budget is increased. For me, an important contribution of the paper is that it explicitly exposes the two situations, or "failure modes", of visit-count-based methods, and SAVE provides improved performance in those situations. Even if SAVE doesn't outperform PUCT with higher search budgets (I don't know if it would?), it could still provide useful intuition for future research that might lead to better performance more generally across wider ranges of search budgets.


Primary comments / questions:

1) Some parts of the paper need more precise language. The text above Eq. 5 discusses the loss in Eq. 5, but does not explicitly reference the equation. The equation just suddenly appears there in between two blocks of text, without any explicit mention of what it contains. After Eq. 6, the paper states that "L_Q may be any variant of Q-learning, such as TD(0) or TD(lambda)". L_Q is a loss function though, whereas Q-learning, TD(0) and TD(lambda) are algorithms, they're not loss functions. I also don't think it's correct to refer to TD(0) and TD(lambda) as "variants of Q-learning". Q-learning is one specific instead of an off-policy temporal difference learning algorithm, TD(lambda) is a family of on-policy temporal difference learning algorithms, and TD(0) is a specific instead of the TD(lambda) family.

2) Why don't the experiments in Figures 2(a-c) include a tabular Q-learner? Since SAVE is, informally, a mix of MCTS and Q-learning, it would be nice to not only compare to MCTS and another MCTS+learning combo, but also standalone Q-learning.

3) The discussion of Tabular Results in 4.1 mentions that the state-value function in PUCT was learned from Monte-Carlo returns. But I think the value function of SAVE was trained using a mix of the standard Q-learning loss and the new amortization loss proposed in the paper. Wouldn't it be more natural to then train PUCT's value function using Q-learning, rather than Monte-Carlo returns?

4) Appendix B.2 mentions that UCT was not required to visit all actions before descending down the tree. I take it this means it's allowed to assign a second visit to a child of the root node, even if some other child does not yet have any visits? What Q-value estimate is used by nodes that have 0 visits? Some of the different schemes I'm aware of would involve setting them to 0, setting them optimistically, setting them pessimistically, or setting them to the average value of the parent. All of these result in different behaviours, and these differences can be especially important in the high-branching-factor / low-search-budget situations considered in this paper.

5) Closely related to the previous point; how does UCT select the action it takes in the "real" environment after completing its search? The standard approach would be to maximise the visit count, but when the search budget is low (perhaps even lower than the branching factor), this can perform very poorly. For example, if every single visit in the search budget led to a poor outcome, it might be preferable to select an unvisited action with an optimistically-initialised Q-value.

6) In 4.2, in the discussion of the Results of Figure 3 (a-c), it is implied that the blue lines depict performance for something that performs search on top of Q-learning? But in the figure it is solely labelled as "Q-learning"? So is it actually something else, or is the discussion text confusing?

7) The discussion of Results in 4.3 mentions that, due to using search, SAVE effectively sees 10 times as many transitions as model-free approaches, and that experiments were conducted on this rather complex Marble Run domain where the model-free approaches were given 10 times as many training steps to correct for this difference. Were experiments in the simpler domains also re-run with such a correction? Would SAVE still outperform model-free approaches in the more simple domains if we corrected for the differences in experience that it gets to see?


Minor Comments (did not impact my score):
- Second paragraph of Introduction discusses "100s or 1000s of model evaluations per action during training, and even upwards of a million simulations per action at test time". Writing "per action" could potentially be misunderstood by readers to refer to the number of legal actions in the root state. Maybe something like "per time step" would have less potential for confusion?
- When I started reading the paper, I was kind of expecting it was going to involve multi-player (adversarial) domains. I think this was because some of the paper's primary motivations involve perceived shortcomings in the Expert Iteration approaches as described by Anthony et al. (2017) and Silver et al. (2018), which were all evaluated in adversarial two-player games. Maybe it would be good to signal at an early point in the paper to the reader that this paper is going to be evaluated on single-agent domains.
- Figure 2 uses red and green, which is a difficult combination of colours for people with one of the most common variants of colour-blindness. It might be useful to use different colours (see https://usabilla.com/blog/how-to-design-for-color-blindness/ for guidelines, or use the "colorblind" palette in seaborn if you use seaborn for plots).
- The error bars in Figure 3 are completely opaque, and overlap a lot. Using transparant, shaded regions could be more easily readable.
- "... model-free approaches because is a combinatorial ..." in 4.2 does not read well.
- Appendix A.3 states that actions were sampled from pi = N / sum N in PUCT. It would be good to clarify whether this was only done when training, or also when evaluating.

**Experience Assessment:**

I have published one or two papers in this area.

**Review Assessment: Checking Correctness Of Derivations And Theory:**

N/A

**Review Assessment: Checking Correctness Of Experiments:**

I assessed the sensibility of the experiments.

**Review Assessment: Thoroughness In Paper Reading:**

I read the paper at least twice and used my best judgement in assessing the paper.

---

> ### Author Response · Authors · 2019-11-07
> **Response to Reviewer #4**
>
> Thank you for your positive review! We believe that SAVE is a potentially important contribution towards improving our collective intuition of hybrid search-and-learning methods and are glad that you find it provides useful insights as well. In fact, you may be interested to know that SAVE came about in part because we tried a PUCT-style approach and were surprised to find that it did not work very well. We think this is an important finding to communicate to the broader research community so—as you said—we can continue to build even stronger agents that work well across a wide range of settings and search budgets.
>
> 1. Thank you for pointing out the imprecise language. We will add a reference to Eq. 5 in the text above and tweak the language to make it flow more naturally so that it doesn’t seem like it appears so suddenly. We will also update the language after Eq. 6 to be more precise.
>
> 2. This is a good point; we should have included a comparison to tabular Q-learning as well. We are working on this and will post a later revision with these results.
>
> 3. We agree that this might make more sense; the reason we train the state-value function from Monte Carlo returns is that this is most similar to the method of training described by PUCT-like methods in the literature such as Silver et al. (2017, 2018). However, we will perform some experiments using Q-learning as well. Our expectation is that this will not help very much, though, as the failure of PUCT is not due to its value function but due to the method by which it learns its policy. Moreover, this is only likely to help in the sparse reward setting, as in the dense reward setting every good action has an immediate positive reward.
>
> 4. Yes, that interpretation is correct. For unvisited actions, we assign a Q-value of 0. In some environments, you are correct that these different ways of setting the Q-values may be better. However, in the Tightrope domain, we think that setting unexplored Q-values to zero is likely the best approach because all possible rewards are greater than or equal to zero. Once an action is found with non-zero reward the best option is to stick with it, so it would not make sense to set the values optimistically. Actions that cause the episode to terminate have a reward of zero, so it would also not make sense to set the values pessimistically as this would lead to over-exploring terminal actions. Setting the values to the average of the parent would either have the effect of setting to zero or setting optimistically (if the parent had positive reward). Thus, it seems to make the most sense to set the unexplored Q-values to zero in this domain.
>
> 5. We select the action to execute in the environment out of those actions which were explored during the search; the specific action that is chosen is the one with the highest Q-value. It is an interesting suggestion to try selecting uniformly at random from the unvisited actions in the case where all estimated values are bad (for example, if they are all zero). We will try an experiment to this effect and update the appendix with the results.
>
> 6. We agree this is confusing and will update both the main text and the appendix to be clearer. Specifically, the “Q-learning” agent is an agent which uses regular Q-learning during training, and at test time additionally performs some amount of search (using the same version of MCTS as that used by SAVE).
>
> 7. In very simple domains like Tightrope, Q-learning may indeed be more efficient in terms of calls to the simulator (e.g., see Figure 2d). However, for moderately complex domains like the Construction tasks, we have seen similar results as with Marble Run: the SAVE agent converges to a higher level of performance that the model-free agent cannot reach even with 10x as much experience. We are working on creating some figures to illustrate this, which we will include in the appendix.
>
> Minor comments: Thank you for these additional comments, we will address these in the text and are working on updating Figures 2 and 3 as per your suggestions.

---

### Comment · Area_Chair1 · 2019-11-13
**Thanks for your reviews. Please take a look at the rebuttal.**

Dear reviewers,

Thank you very much for your efforts in reviewing this paper.

The authors have provided their rebuttal. It would be great if you take a look at them, and see whether it changes your opinion in anyway. If there is still any unclear point or a serious disagreement, please bring it up. Also if you are hoping to see a specific change or clarification in the paper before you update your score, please mention it.

The authors have only until November 15th to reply back.

I also encourage you to take a look at each others’ reviews. There might be a remark in other reviews that changes your opinion.

Thank you,
Area Chair

---

> ### Comment · AnonReviewer4 · 2019-11-14
> **I have looked at the rebuttals.**
>
> I confirm that I have read the authors' responses and other reviews. The authors' responses are satisfactory for me. I see no need to update my score (which was already positive).
>
> If possible, it would be nice to already see an updated version of the paper with some of the simpler updates as discussed in various reviews and responses (like clarifications in various parts of the text). Since the authors' responses leave the impression to me that they understand where our confusion is coming from in various points raised in the reviews, and have promised to address these issues, I trust that they will be able to do this successfully. So, seeing these updates on/before November 15 is not crucial to me -- it would just be nice to see already if possible.
>
> I understand that already including updates that necessitate the running of additional experiments may be infeasible in a short amount of time, if the experiments are still in progress.

---

### Author Response · Authors · 2019-11-14
**Updated version (1/2)**

Dear reviewers,

We have now made the additional changes as promised, which we feel have improved the paper—thank you for the suggestions! We hope we have been able to address all of your concerns, but welcome additional feedback if you feel there is more we can do to strengthen the paper.

We note that while performing the additional experiments in tabular Tightrope (in response to R4), we performed a new hyperparameter scan and found that in the dense setting of Tightrope a smaller setting of the UCT constant (i.e., making the search greedier) increased the performance of the PUCT baseline. However, the overall pattern of results remains the same in the dense setting, and the quantitative results in the sparse reward setting (which is more representative of the rest of environments) stayed exactly the same: PUCT overall performs less well in cases with sparser rewards and smaller search budgets. We have updated Figure 2 with these results. We find it interesting that in all our experiments SAVE seems to be relatively robust to the setting of c, while other MCTS methods like PUCT are much more sensitive to this parameter.

Experiments:

- R2, R3, R4: We have included experiments with tabular Q-learning in Figure 2. We find that tabular SAVE outperforms tabular Q-learning in all of our Tightrope experiments. Tabular PUCT can outperform Q-learning when given a higher search budget, though tends to underperform Q-learning with lower search budgets. We have additionally added some discussion of this to the main text.

- R2: We have started experiments on the Covering task using several different exploration types: epsilon-greedy over estimated Q-values, categorical sampling from the softmax of estimated Q-values, categorical sampling from the normalized visit counts, and UCB. We find that using epsilon-greedy (which is what we were using previously) works the best out of these exploration strategies by a substantial margin. We speculate that this may be because it is important for the Q-function to be well approximated across all actions, so that it is useful during MCTS backups. However, UCB and categorical methods do not uniformly sample the action space, meaning that some actions are very unlikely to be ever learned from. The amortization loss does not help either, as these actions will not be explored during search either. The error in the Q-values for unexplored actions grows over time (due to catastrophic forgetting), leading to a poorly approximated Q-function that is unreliable. In contrast, epsilon-greedy consistently spends a little bit of time exploring these actions, preventing their values from becoming too inaccurate. We hypothesize this might be less of a problem if we were to use a separate state-value function for bootstrapping (as is done by AlphaZero), which we plan to explore in future work. We have added the current results of these experiments, and this discussion, to the appendix (see Figure C.3 and Section C.4). We will additionally update the figure with the final results once training is complete.

- R4: We experimented with using one-step Q-learning to learn an action-value function in the tabular PUCT agent, rather than using Monte Carlo returns to learn a state-value function, and find that these two approaches result in similar levels of performance.

- R4: We experimented with an action selection policy for the UCT agent which chooses an action at random from unvisited actions if all of the explored actions have an expected value of zero (which is the expected value of bad/terminal actions). We find that this indeed improves performance. While the effect is statistically significant (p=0.02) the effect size is quite small: on the dense setting with M=95% we achieve a median reward of 0.08 (using this thresholding action selection policy) versus 0.07 (selecting the max of visited actions). We have added these results and discussion to the appendix (Section B.2).

- R2, R4: We have included results on Covering showing that SAVE results in higher performance than Q-learning, even when controlling for the same number of environment interactions. We have both mentioned this result in the main text, and included a new Figure C.4 in the appendix.

---

> ### Author Response · Authors · 2019-11-14
> **Updated version (2/2)**
>
> Text and figures:
>
> - R2, R4: We have clarified in the text what the “Q-Learning” agent is, and what it means for it to have a “test budget” in Figure 3.
>
> - R2, R3: We have clarified the difference between past work and our contributions, and provided further justification for SAVE’s approach.
>
> - R2: We have added a comment in the main text about including the temperature parameter in the softmax function.
>
> - R2: We have clarified how the MDP works in Tightrope.
>
> - R2: We have clarified that SAVE and PUCT may both perform well with larger search budgets, but that our emphasis is on the regime with small search budgets.
>
> - R2: We have added discussion about the performance of PUCT in the Construction tasks.
>
> - R4: We have made the explanation around Eq. 5 flow more naturally and used more precise language around Eq. 6.
>
> - R4: We have updated the figures to use a colorblind-friendly palette, and we have updated the error bars in the figures to have caps, to make them easier to parse.
>
> - R2, R3, R4: We have updated the text based on all of the additional smaller comments as well.

---

### Decision · Program_Chairs · 2019-12-19

**Decision:**

Accept (Poster)

**Comment:**

This paper proposes Search with Amortized Value Estimates (SAVE) that combines Q-learning and MCTS.  SAVE uses the estimated Q-values obtained by MCTS at the root node to update the value network, and uses the learned value function to guide MCTS.

The rebuttal addressed the reviewers’ concerns, and they are now all positive about the paper. I recommend acceptance.